# Is Generation Required for Data-Efficient Perception?

**Jack Brady** [1 2 3] **Bernhard Schölkopf** [1 2 3] **Thomas Kipf** [4] **Simon Buchholz** [* 1 2] **Wieland Brendel** [* 1 2 3]

## Abstract

It has been hypothesized that achieving the data efficiency of human visual perception requires a *generative* approach in which internal representations result from inverting a *decoder*. Yet today's most successful vision models are *non-generative*, relying on an *encoder* that maps images to representations without decoder inversion. This raises the question of whether generation is necessary for data-efficient machine perception. To address this, we study to what extent generative and non-generative methods can achieve *compositional generalization*, a hallmark of human data efficiency. Under a compositional generative process, we formally characterize the inductive biases required for compositional generalization in decoder-based (generative) and encoder-based (non-generative) methods. We show theoretically that the inductive biases required for an encoder are substantially more complex and generally infeasible to impose explicitly through architectural constraints or regularization. By contrast, the decoder biases take a simple form that can be enforced directly. These results suggest that compositional generalization may be substantially easier to achieve through a generative paradigm of learning and inverting a decoder rather than learning an encoder directly. We examine our theoretical findings empirically by training a range of generative and non-generative methods on synthetic image data. We find that non-generative methods often fail to generalize compositionally and require large-scale pretraining to improve generalization. By comparison, generative methods yield gains in generalization without requiring additional data.

## 1. Introduction

Perceiving the visual world requires an internal representation of visual input. Two opposing views exist for how such representations should be constructed. The **generative** view posits that representations result from inverting a *decoder* to recover the latent variables that generate an image (von Helmholtz, 1867; Friston & Stephan, 2007; Hinton, 2007; Olshausen, 2014). Conversely, the **non-generative** view holds that representations result from an *encoder* that directly maps images to latent variables without inverting a decoder (Yamins et al., 2014; LeCun, 2022; Gibson, 1979). A core problem in AI is to understand which of these paradigms should be adopted to build machines with *human-level visual perception*.

In recent years, consensus around this problem has shifted, following breakthroughs in non-generative methods for representation learning (Radford et al., 2021; Caron et al., 2021; Tschannen et al., 2025; Oquab et al., 2024). These methods, trained with self- or weak supervision, now enable unprecedented performance on perceptual tasks such as object recognition (Siméoni et al., 2025) and image captioning (Fan et al., 2025; Beyer et al., 2024). This progress has given rise to a common assumption that non-generative methods provide the most promising path toward human-level visual perception, while generative approaches are not necessary (Balestriero & LeCun, 2024).

Yet, despite their remarkable performance, current non-generative methods fall short in another key pillar of human visual perception: *data efficiency*. Specifically, these methods rely on web-scale datasets in which different visual concepts are encountered across diverse contexts, with high frequency (Udandarao et al., 2024), and often with language supervision (Zhang et al., 2024). In contrast, human children achieve robust visual perception through much more constrained data, encountering concepts only a handful of times, mainly in the same settings (e.g., the home), and with little supervision (Tenenbaum et al., 2011; Lake et al., 2017). To reach this level of data efficiency, it has been conjectured across several disciplines (Lake et al., 2015; Kilbertus et al., 2018; Peters et al., 2024) that a generative approach may be necessary. This raises a key question: Can non-generative approaches to perception also achieve human-level data efficiency, or is generation required?

---

[*]Shared last author [1]Max Planck Institute for Intelligent Systems, Tübingen, Germany [2]Tübingen AI Center [3]ELLIS Institute, Tübingen [4]Google Deepmind. Correspondence to: Jack Brady, Simon Buchholz <first.last@tue.mpg.de>.

*Proceedings of the 43$^{rd}$ International Conference on Machine Learning*, Seoul, South Korea. PMLR 306, 2026. Copyright 2026 by the author(s).

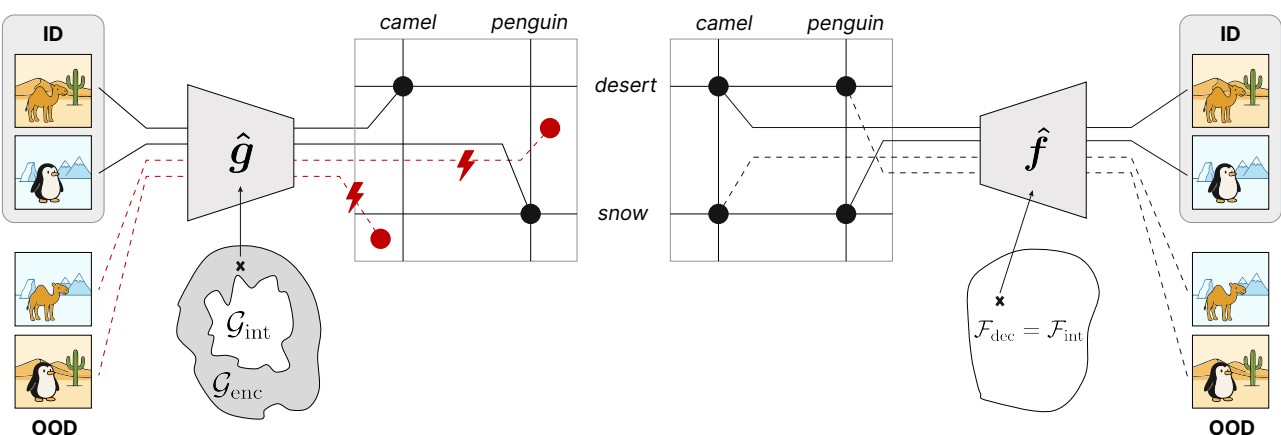

*Figure 1.* **Generative vs. non-generative compositional generalization.** We assume in-domain (ID) and out-of-domain (OOD) images arise from a latent variable model through an unknown compositional generator $\boldsymbol{f} \in \mathcal{F}_{\text{int}}$, with inverse $\boldsymbol{g} \in \mathcal{G}_{\text{int}}$. In this setting, achieving compositional generalization for a generative approach requires inductive biases on a *decoder* such that $\hat{\boldsymbol{f}} \in \mathcal{F}_{\text{int}}$, and for a non-generative approach, an encoder such that $\hat{\boldsymbol{g}} \in \mathcal{G}_{\text{int}}$ (Sec. 2). We show theoretically in (Sec. 3) that imposing such biases on a decoder is much simpler than for an encoder. Empirically, this tends to manifest in an encoder yielding incorrect representations for OOD images (Sec. 5.1). In contrast, a decoder is able to correctly generate such images enabling compositional generalization through inversion (Sec. 4, 5.2).

In this work, we approach this question by analyzing, both in theory and practice, to what extent generative methods are required for *compositional generalization*, a key mechanism underlying human data efficiency (Fodor & Pylyshyn, 1988; Greff et al., 2020). Compositional generalization is the ability to perceive out-of-domain (OOD) scenes containing unseen combinations of concepts, e.g., a dog in the park after only seeing dogs in the house. Since these scenes are absent from the training distribution, achieving such generalization requires imposing inductive biases on a model that induce correct OOD behavior. Thus, we recast the question of whether generative methods are necessary for human data efficiency as one of inductive bias: What biases are required for compositional generalization, and how feasibly can they be imposed in generative versus non-generative methods?

**Theoretical Analysis.** We first investigate this question theoretically by assuming that images are generated by a compositional latent variable model (Brady et al., 2025) (Sec. 2). Within this setting, we can precisely characterize the inductive biases required for compositional generalization in generative (decoder-based) and non-generative (encoder-based) methods. We show that while both classes of methods can, in principle, achieve compositional generalization, the required biases differ substantially in complexity. For generative methods, the inductive biases are data-independent and always take the form of a simple block-diagonal structure on the derivative matrix of the decoder. In contrast, for non-generative methods, the corresponding encoder structure is defined only relative to the tangent space of the observed data manifold. Because this tangent space depends on the geometry of the data and is not known outside the training distribution, the required encoder biases are inherently

data-dependent, substantially more complex, and ill-posed to enforce explicitly OOD. These results suggest that learning and inverting a decoder may provide a much simpler route to compositional generalization than imposing the corresponding inductive biases directly on an encoder.

**Empirical Analysis.** To investigate these findings empirically, we first show how decoder inversion can be performed efficiently both in-domain (ID), via an autoencoder, and out-of-domain (OOD), via gradient-based search (Sec. 4.1) and generative replay (Sec. 4.2). In Sec. 5, we then evaluate a range of generative and non-generative methods on photorealistic synthetic image datasets (Bordes et al., 2023) across different ID and OOD splits. We find that, across a variety of inductive biases, non-generative methods generally exhibit poor OOD performance when trained from scratch and improvements require larger-scale pretraining. By comparison, generative methods can achieve significant improvements OOD, without requiring additional data, through inductive biases on a decoder along with search and replay.

## 2. Problem Setup

In this section, we first provide formal definitions for generative and non-generative perception, as well as compositional generalization. We then define a structured latent variable model that allows us to precisely characterize the inductive biases required for compositional generalization in both classes of methods.

**Perception.** We begin by formalizing *visual perception*. To this end, we assume that images $\boldsymbol{x} \in \mathcal{X} \subset \mathbb{R}^{d_x}$ arise from a latent variable model. Specifically, we assume $\boldsymbol{x}$ is generated from a latent vector $\boldsymbol{z} \in \mathcal{Z} := \mathbb{R}^{d_z}$ by a diffeomorphic

generator $\boldsymbol{f} : \mathcal{Z} \to \mathcal{X}$, i.e., $\boldsymbol{x} = \boldsymbol{f}(\boldsymbol{z})$. Visual concepts in $\boldsymbol{x}$ (e.g. "camel" and "desert" in Fig. 1) are modelled as $K$ distinct *slots* of latents $\boldsymbol{z}_k \in \mathbb{R}^m$ such that $\boldsymbol{z} = (\boldsymbol{z}_1, ..., \boldsymbol{z}_K)$ (Brady et al., 2025). Now, assume we have a representation of an image $\hat{\boldsymbol{z}} = \phi(\boldsymbol{x})$, where $\phi : \mathbb{R}^{d_x} \to \mathcal{Z}$. We define perception as the ability to invert the generator $\boldsymbol{f}$ via $\phi$ to recover the slots $\boldsymbol{z}_k$ that generated $\boldsymbol{x}$. In general, recovering $\boldsymbol{z}_k$ exactly is impossible. Thus, we only require that $\phi$ inverts $\boldsymbol{f}$ up to re-parameterizations and permutations of the slots. Formally, let $\boldsymbol{h}_\pi$ be a function composed of slot-wise bijections $\boldsymbol{h}_k : \mathbb{R}^m \to \mathbb{R}^m$ and permutations $\pi$, i.e., $\boldsymbol{h}_\pi(\boldsymbol{z}) := \{\boldsymbol{h}_k(\boldsymbol{z}_{\pi(k)})\}_{k=1}^K$. Perception on a set $\mathcal{Z}^S \subseteq \mathcal{Z}$ requires that there exist an $\boldsymbol{h}_\pi$ such that

$$\forall \boldsymbol{z} \in \mathcal{Z}^S, \;\; \phi(\boldsymbol{f}(\boldsymbol{z})) = \boldsymbol{h}_\pi(\boldsymbol{z}). \tag{2.1}$$

Eq. (2.1) takes the perspective of perception as an inverse problem (Tenenbaum et al., 2011), but with respect to the ground-truth generator $\boldsymbol{f}$. This contrasts with a task-based view (Yamins & DiCarlo, 2016) where perception is defined with respect to solving a downstream task. We note that the task-based view can be framed as a special case of Eq. (2.1), by treating task-specific predictions as the latent variables to be recovered by $\phi$. Moreover, if a representation satisfying Eq. (2.1) is learned, downstream tasks such as object classification can be solved via a simple readout applied independently to each inferred slot $\hat{\boldsymbol{z}}_k$ (see Sec. 5).

**Generative and non-generative approaches.** We now characterize the *generative* and *non-generative* approaches to perception. For the generative approach, representations are obtained by inverting a learned *decoder* $\hat{\boldsymbol{f}} : \mathcal{Z} \to \mathbb{R}^{d_x}$, i.e., $\phi(\boldsymbol{x}) = \hat{\boldsymbol{f}}^{-1}(\boldsymbol{x})$. For this to satisfy Eq. (2.1), the decoder $\hat{\boldsymbol{f}}$ must *identify* the ground-truth generator $\boldsymbol{f}$ such that for $\boldsymbol{z} \in \mathcal{Z}$

$$\hat{\boldsymbol{f}}(\boldsymbol{h}_\pi(\boldsymbol{z})) = \boldsymbol{f}(\boldsymbol{z}). \tag{2.2}$$

Alternatively, for the non-generative approach, a representation is defined as $\phi(\boldsymbol{x}) = \hat{\boldsymbol{g}}(\boldsymbol{x})$, where $\hat{\boldsymbol{g}} : \mathbb{R}^{d_x} \to \mathcal{Z}$ is a learned *encoder* which *is not* constructed to invert a decoder $\hat{\boldsymbol{f}}$. For this to satisfy Eq. (2.1), $\hat{\boldsymbol{g}}$ must identify the inverse generator $\boldsymbol{g} := \boldsymbol{f}^{-1}$ such that for $\boldsymbol{x} \in \mathcal{X}$

$$\hat{\boldsymbol{g}}(\boldsymbol{x}) = \boldsymbol{h}_\pi(\boldsymbol{g}(\boldsymbol{x})). \tag{2.3}$$

We emphasize that the difference between generative and non-generative approaches is not whether an encoder or decoder is used. Instead, it is whether a representation satisfying Eq. (2.1) is obtained by learning an approximation $\hat{\boldsymbol{f}}$ of the ground-truth generator (Eq. (2.2)) and then inverting this model, or by learning an approximation $\hat{\boldsymbol{g}}$ of the inverse generator directly (Eq. (2.3)).

**Compositional generalization.** We now formalize *compositional generalization*. Informally, compositional generalization is the ability to perceive OOD images containing unseen concept combinations (e.g. "penguin" and "desert"

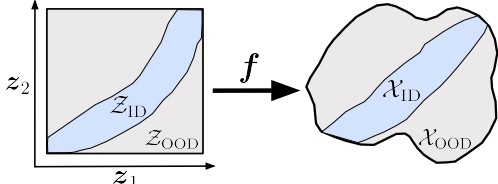

*Figure 2.* Visualization of a data generating process with in- and out-of-domain regions.

in Fig. 1). To formalize this, we assume observed images $\mathcal{X}_{\text{ID}} \subset \mathcal{X}$ arise from only a subset of possible concept combinations $\mathcal{Z}_{\text{ID}} \subset \mathcal{Z}$, i.e., $\mathcal{X}_{\text{ID}} := \boldsymbol{f}(\mathcal{Z}_{\text{ID}})$ (see Fig. 2). OOD concept combinations $\mathcal{Z}_{\text{OOD}}$ are defined as the set of all unseen combinations of slots

$$\mathcal{Z}_{\text{OOD}} := \{\mathcal{Z}_1 \times \mathcal{Z}_2 \times \cdots \times \mathcal{Z}_K\} \setminus \mathcal{Z}_{\text{ID}} \quad \text{with} \\ \mathcal{Z}_k := \{\boldsymbol{z}_k \in \mathbb{R}^m \mid \boldsymbol{z} \in \mathcal{Z}_{\text{ID}}\}, \tag{2.4}$$

which give rise to OOD images $\mathcal{X}_{\text{OOD}} := \boldsymbol{f}(\mathcal{Z}_{\text{OOD}})$ (Fig. 2). Assuming Eq. (2.1) is satisfied on $\mathcal{Z}_{\text{ID}}$, compositional generalization is achieved if Eq. (2.1) it is also satisfied OOD, for all $\boldsymbol{z} \in \mathcal{Z}_{\text{OOD}}$.

**The problem of OOD identifiability.** Without restrictions on the ground-truth generator $\boldsymbol{f}$, compositional generalization, using a generative or non-generative approach, is an ill-posed problem. Specifically, consider two generators $\boldsymbol{f}^1, \boldsymbol{f}^2 \in \mathcal{F}$ such that on $\mathcal{Z}_{\text{ID}}$, $\boldsymbol{f}^1 \circ \boldsymbol{h}_\pi = \boldsymbol{f}^2$, but on $\mathcal{Z}_{\text{OOD}}$, $\boldsymbol{f}^1 \circ \boldsymbol{h}_\pi \neq \boldsymbol{f}^2$. In this case, identifying the ground-truth $\boldsymbol{f}$ or its inverse, out-of-domain, is impossible since there is no way to distinguish these generators from in-domain data. Thus, for compositional generalization to be possible, $\boldsymbol{f}$ must belong to a function class $\mathcal{F}$ such that for all $\boldsymbol{f}^1, \boldsymbol{f}^2 \in \mathcal{F}$

$$\forall \boldsymbol{z} \in \mathcal{Z}_{\text{ID}}, \; \boldsymbol{f}^1(\boldsymbol{h}_\pi(\boldsymbol{z})) = \boldsymbol{f}^2(\boldsymbol{z}) \\ \implies \forall \boldsymbol{z} \in \mathcal{Z}_{\text{OOD}}, \; \boldsymbol{f}^1(\boldsymbol{h}_\pi(\boldsymbol{z})) = \boldsymbol{f}^2(\boldsymbol{z}), \tag{2.5}$$

which equivalently implies that for all inverses $\boldsymbol{g}^1, \boldsymbol{g}^2 \in \mathcal{G} := \{\boldsymbol{f}^{-1} \mid \boldsymbol{f} \in \mathcal{F}\}$

$$\forall \boldsymbol{x} \in \mathcal{X}_{\text{ID}}, \; \boldsymbol{h}_\pi(\boldsymbol{g}^1(\boldsymbol{x})) = \boldsymbol{g}^2(\boldsymbol{x}) \\ \implies \forall \boldsymbol{x} \in \mathcal{X}_{\text{OOD}}, \; \boldsymbol{h}_\pi(\boldsymbol{g}^1(\boldsymbol{x})) = \boldsymbol{g}^2(\boldsymbol{x}). \tag{2.6}$$

**Further assumptions on $\boldsymbol{f}$.** Under our current assumptions, $\mathcal{F}$ consist of all diffeomorphisms from $\mathcal{Z}$ to $\mathcal{X}$. This function class is far too large to satisfy Eq. (2.5). Thus, further assumptions are required. Recently, Brady et al. (2025, Thm. 4.4) proved that diffeomorphisms (on their image) with the following form will satisfy Eq. (2.5)

$$\boldsymbol{f}(\boldsymbol{z}) = \sum_{k=1}^K \boldsymbol{f}^k(\boldsymbol{z}_k) + \sum_{\boldsymbol{\alpha} : |\boldsymbol{\alpha}| \leq n} \boldsymbol{c}_{\boldsymbol{\alpha}} \boldsymbol{z}^{\boldsymbol{\alpha}}, \tag{2.7}$$

where $n \in \mathbb{N}$, $\boldsymbol{c_\alpha} \in \mathbb{R}^{d_x}$, and $\boldsymbol{\alpha} \in \mathbb{N}_0^{d_z}$ is a *multi-index*.[1] This function class, denoted $\mathcal{F}_{\text{int}}$, was introduced to model concepts with varying degrees of interaction $n$. For example, when $n = 1$, the second-sum on the RHS vanishes and concepts can only interact additively (Lachapelle et al., 2023). For $n > 1$ concepts can interact explicitly via polynomial functions of components from different slots. This aims to capture more complex concept interactions such as between objects and backgrounds. Such functions thus offer a flexible model of visual concepts, and are the largest function class shown to satisfy (Eq. (2.5)). For these reasons, we assume that ground-truth generators $\boldsymbol{f}$ belong to $\mathcal{F}_{\text{int}}$, and inverse generators $\boldsymbol{g}$ to $\mathcal{G}_{\text{int}} := \{\boldsymbol{f}^{-1} \mid \boldsymbol{f} \in \mathcal{F}_{\text{int}}\}$.

**Inductive biases for compositional generalization.** Under this generative process, we can now formalize what is required to guarantee compositional generalization using both a generative and non-generative approach. To this end, we assume that Equations 2.2 and 2.3 are satisfied in-domain, for a decoder $\hat{\boldsymbol{f}}$ and encoder $\hat{\boldsymbol{g}}$, respectively. Since ground-truth generators $\mathcal{F}_{\text{int}}$ and inverses $\mathcal{G}_{\text{int}}$ satisfy Equations 2.5 and 2.6, compositional generalization is guaranteed in theory by restricting the decoder class out-of-domain such that $\hat{\boldsymbol{f}} \in \mathcal{F}_{\text{int}}$ and, similarly, the encoder class such that $\hat{\boldsymbol{g}} \in \mathcal{G}_{\text{int}}$. In practice, however, such restrictions on a model's OOD behavior must come from either explicit inductive biases such as architectural design or regularization or implicit biases such optimization dynamics. Whether it is possible to impose such inductive biases on a decoder to ensure $\hat{\boldsymbol{f}} \in \mathcal{F}_{\text{int}}$ or on an encoder to ensure $\hat{\boldsymbol{g}} \in \mathcal{G}_{\text{int}}$ depends critically on the structural properties of these ground-truth function classes.

# 3. Theoretical Analysis

In this section, we theoretically analyze the structure of $\mathcal{F}_{\text{int}}$ and $\mathcal{G}_{\text{int}}$ to understand whether a model can be constrained to these classes with explicit or implicit inductive biases.

**Structure of $\mathcal{F}_{\text{int}}$.** Generators in $\mathcal{F}_{\text{int}}$ are defined as diffeomorphisms which take the form of Eq. (2.7). Consequently, to enforce $\hat{\boldsymbol{f}} \in \mathcal{F}_{\text{int}}$, we must constrain a decoder to match this form. This can be done in a straightforward manner via architecture design. For example, the first term on the RHS of Eq. (2.7) can be parameterized as the sum of slot-wise neural networks and the second term using learned coefficients for $\boldsymbol{c_\alpha}$. Furthermore, we highlight that functions of the form in Eq. (2.7) can equivalently be expressed as having block-diagonal derivative matrices $D^{n+1}\boldsymbol{f}(\boldsymbol{z})$ (Brady et al., 2025; Lachapelle et al., 2023). Specifically, if $n = 1$, then the Hessian $D^2\boldsymbol{f}$ has the structure that for any two

---

[1]A *multi-index* is an ordered tuple $\boldsymbol{\alpha} = (\alpha_1, \alpha_2, ..., \alpha_d)$ of non-negative integers $\alpha_i \in \mathbb{N}_0$, with operations $|\boldsymbol{\alpha}| := \alpha_1 + \alpha_2 + ... + \alpha_d$, and $\boldsymbol{z^\alpha} := z_1^{\alpha_1} z_2^{\alpha_2} ... z_d^{\alpha_d}$.

slots $\boldsymbol{z}_k$ and $\boldsymbol{z}_l$,

$$\forall 1 \le k \ne l \le K, \quad D_{\boldsymbol{z}_k} D_{\boldsymbol{z}_l} \boldsymbol{f}(\boldsymbol{z}) = 0. \quad (3.1)$$

For $n > 1$, analogous conditions hold for higher-order derivatives (Brady et al., 2025). Thus, we can also enforce that $\hat{\boldsymbol{f}} \in \mathcal{F}_{\text{int}}$ for a decoder $\hat{\boldsymbol{f}}$ via regularization. For example, when $n = 1$, we can use the following regularizer (with similar expressions for higher-order derivatives when $n > 1$)

$$\mathcal{R}(\hat{\boldsymbol{f}}, \boldsymbol{z}) = \sum_{k \ne l \in [K]} \left\| D^2_{\boldsymbol{z}_k, \boldsymbol{z}_l} \hat{\boldsymbol{f}}(\boldsymbol{z}) \right\|. \quad (3.2)$$

### 3.1. Structure of $\mathcal{G}_{\text{int}}$.

We now investigate the structure of inverse generators $\mathcal{G}_{\text{int}}$. Our goal is to characterize these functions in terms of their derivative structure, analogous to our characterization of $\mathcal{F}_{\text{int}}$. This is more challenging, however, since inverse generators $\boldsymbol{g} \in \mathcal{G}_{\text{int}}$ do not admit an analytical form similar to Eq. (2.7). As a result, understanding their derivative structure requires a more detailed analysis. For simplicity, we present results for $n = 1$; similar statements can in principle be derived for higher order derivatives for the case $n > 1$. We also study whether we can find architectures with an inductive bias towards $\mathcal{G}_{\text{int}}$, but delegate this to Appendix A.2.

We will first assume that the observed dimension $d_x$ equals the ground-truth latent dimension $d_z$ such that $\mathcal{X} = \mathcal{Z}$. In this case, we show that, similar to generators in $\mathcal{F}_{\text{int}}$, inverse generators in $\mathcal{G}_{\text{int}}$ have a structured Jacobian and Hessian. Specifically, we prove the following result.

**Lemma 3.1.** *Let $\boldsymbol{g} \in \mathcal{G}_{\text{int}}$ for $n = m = 1$ and $d_x = d_z$. Then $\boldsymbol{g}$ has the property that for $\boldsymbol{x} \in \mathcal{X}$*

$$(D\boldsymbol{g})^{-\top}(\boldsymbol{x})D^2\boldsymbol{g}_s(\boldsymbol{x})(D\boldsymbol{g})^{-1}(\boldsymbol{x}) \in \text{Diag}(d_x) \quad (3.3)$$

*is a diagonal matrix for $s \in [d_z]$. Further, if $\boldsymbol{g}$ is a diffeomorphism satisfying Eq. (3.3) then $\boldsymbol{g} \in \mathcal{G}_{\text{int}}$.*

Thus, when $\mathcal{X} = \mathcal{Z}$, enforcing that $\hat{\boldsymbol{g}} \in \mathcal{G}_{\text{int}}$ requires constraining an encoder according to Eq. (3.3). This is achievable, for instance, through regularization on the derivatives of $\hat{\boldsymbol{g}}$ analogous to Eq. (3.2).

This setting, however, is not applicable to image data since images typically lie in a manifold embedded in a higher-dimensional ambient space $\mathbb{R}^{d_x}$. We therefore consider the more practical case where $d_x$ is larger than the ground-truth latent dimension $d_z$. Specifically, we assume $d_x \ge d_z^3$. In this case, we first prove that the aforementioned structure on $D\boldsymbol{g}$ and $D^2\boldsymbol{g}$ is no longer present.

**Theorem 3.2.** *Assume that $d_x \ge d_z^3$. Let $\boldsymbol{B}_l \in \mathbb{R}^{d_x \times d_x}$ be symmetric matrices for $1 \le l \le d_z$. Then there is for any $\boldsymbol{x}_0 \in \mathbb{R}^{d_x}$ and for almost every $\boldsymbol{A} \in \mathbb{R}^{d_z \times d_x}$ a generator $\boldsymbol{f} \in \mathcal{F}_{\text{int}}$ with a (left)-inverse $\boldsymbol{g} \in \mathcal{G}_{\text{int}}$, such that $\boldsymbol{f}(0) = \boldsymbol{x}_0$ and $D\boldsymbol{g}(\boldsymbol{x}_0) = \boldsymbol{A}$ and $D^2\boldsymbol{g}_l(\boldsymbol{x}_0) = \boldsymbol{B}_l$ for $1 \le l \le d_z$.*

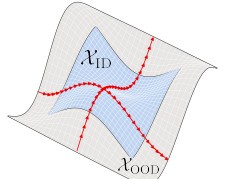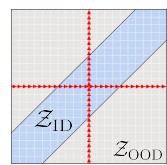

*Figure 3.* Structure of a data manifold $\mathcal{X}$ and latent manifold $\mathcal{Z}$.

Thus, when $d_x \gg d_z$, $D^2 \boldsymbol{g}$ and $D\boldsymbol{g}$ can be arbitrary matrices (up to a set of measure 0). We emphasize that this result applies to $\mathcal{F}_{\text{int}}$ with arbitrary interaction degree $n \geq 1$ and any slot dimensions. However, the structure expressed in Eq. (3.3) does not vanish entirely from $\boldsymbol{g}$. Instead, it persists, but only for the restriction of $\boldsymbol{g}$ to the data manifold $\mathcal{X}$. Specifically, the constraint Eq. (3.3) holds more generally for $n = m = 1$ when $D\boldsymbol{g}$ is projected on the tangent space $T_{\boldsymbol{x}}\mathcal{X}$ of the data manifold, i.e.,

$$\left( (D\boldsymbol{g}(\boldsymbol{x})\Pi_{T_{\boldsymbol{x}}\mathcal{X}})^+ \right)^\top D^2 \boldsymbol{g}_s(\boldsymbol{x}) \, (D\boldsymbol{g}(\boldsymbol{x})\Pi_{T_{\boldsymbol{x}}\mathcal{X}})^+ \in \text{Diag}(d_z) \tag{3.4}$$

where $\Pi_{T_{\boldsymbol{x}}\mathcal{X}}$ denotes the orthogonal projection on the tangent space (see Lemma A.5 for details).

Constraining an encoder such that $\hat{\boldsymbol{g}} \in \mathcal{G}_{\text{int}}$ thus requires enforcing this structure on $\hat{\boldsymbol{g}}$. This is challenging because the constraints depend on the geometry of the data manifold $\mathcal{X}$. Enforcing such constraints is thus ill-posed since the geometry of out-of-domain regions $\mathcal{X}_{\text{OOD}} \subset \mathcal{X}$ is unobserved. This suggests that constraining an encoder through approaches such as architectural design or regularization is infeasible, as any such method would necessarily be data-dependent as well as implicitly assume knowledge of $\mathcal{X}_{\text{OOD}}$.

We contrast this with the reverse direction for $\boldsymbol{f} \in \mathcal{F}_{\text{int}}$. In this case, the structure to be enforced (see Eq. (3.1)) is not manifold-dependent but is always aligned with the global coordinate axes (Fig. 3, right). This allows for a universal procedure to constrain a decoder to $\mathcal{F}_{\text{int}}$, rather than a manifold-dependent one (Fig. 3, left). Moreover, such constraints can also be applied in OOD regions, since the manifold $\mathcal{Z}_{\text{ID}}$ extends in a Cartesian fashion and its structure is therefore known.

**Special case of $n = 0$.** We briefly discuss the case of functions in $\mathcal{F}_{\text{int}}$ when $n = 0$. These functions, introduced by Brady et al. (2023), are a special case of $n = 1$ with the additional, more restrictive condition $|D_{\boldsymbol{z}_k} \boldsymbol{f}_i(\boldsymbol{z})| \cdot |D_{\boldsymbol{z}_l} \boldsymbol{f}_i(\boldsymbol{z})| = 0$ for each $i \in [d_x]$. In other words, each pixel $i$ depends only on a single slot and no interactions (such as occlusions) between objects are possible. In this case, we can find a left inverse $\boldsymbol{g}$ of $\boldsymbol{f} \in \mathcal{F}_{\text{int}}^{n=0}$ (for any $d_x \geq d_z$) whose Jacobian satisfies the sparsity constraint $|D_{\boldsymbol{x}_i} \boldsymbol{g}_k| \cdot |D_{\boldsymbol{x}_j} \boldsymbol{g}_l| = 0$ for $l \neq k$. This additional

structure can thus be leveraged to restrict $\mathcal{G}_{\text{enc}}$. However, this remains challenging in practice because the sparsity pattern (i.e., which slots $\boldsymbol{z}_l$ depends on which pixel $\boldsymbol{x}_i$) is not known a-priori. In Section 5, we study whether concepts satisfying $n = 0$ can empirically enable compositional generalization for non-generative approaches.

**Takeaways.** Our results suggest that it is generally not feasible to constrain an encoder such that $\hat{\boldsymbol{g}} \in \mathcal{G}_{\text{int}}$. This means that there can exist encoders $\hat{\boldsymbol{g}} \in \mathcal{G}_{\text{enc}}$ that satisfy Eq. (2.3) ID, but fail to generalize this behavior OOD (see Fig. 1). In practice, this does not necessarily imply that optimization will always recover such an encoder. However, we lack a principled means to prevent this. Thus, for non-generative methods, whether compositional generalization occurs depends on whether optimization happens to avoid converging to encoders that do not generalize OOD. In contrast, generative methods can avoid such solutions by construction, through appropriate inductive biases on a decoder. We investigate the impact of this asymmetry empirically in Sec. 5.

## 4. Search and Replay

Our results in Sec. 3 suggest that ensuring compositional generalization through inductive biases requires a generative approach, i.e., inverting a learned decoder $\hat{\boldsymbol{f}}$. If a decoder admits an explicit inverse, this inversion is trivial. For image data, however, constructing such a decoder is challenging as this generally requires that $\mathcal{X} = \mathbb{R}^{d_x}$ (Papamakarios et al., 2021). Consequently, inverting $\hat{\boldsymbol{f}}$ requires solving an inference problem: given an image $\boldsymbol{x}$, we must find a latent $\boldsymbol{z}^*$ such that

$$\boldsymbol{x} = \hat{\boldsymbol{f}}(\boldsymbol{z}^*). \tag{4.1}$$

In this section, we explore strategies for solving this inference problem efficiently.

**Inversion on $\mathcal{X}_{\text{ID}}$.** For in-domain images, i.e. $\boldsymbol{x} \in \mathcal{X}_{\text{ID}}$, inverting a decoder $\hat{\boldsymbol{f}}$ to obtain $\boldsymbol{z}^*$ can be done directly by training an *autoencoder*. Specifically, we can leverage an encoder $\hat{\boldsymbol{g}}$ to invert $\hat{\boldsymbol{f}}$ in-domain by minimizing the reconstruction objective

$$\min_{\hat{\boldsymbol{f}}, \hat{\boldsymbol{g}}} \mathbb{E}_{\boldsymbol{x} \sim \mathcal{X}_{\text{ID}}} \left\| \boldsymbol{x} - \hat{\boldsymbol{f}}(\hat{\boldsymbol{g}}(\boldsymbol{x})) \right\|^2. \tag{4.2}$$

Thus, for images $\boldsymbol{x} \in \mathcal{X}_{\text{ID}}$, $\boldsymbol{z}^*$ (Eq. (4.1)) can be obtained directly as the output of the encoder. For out-of-domain images, however, minimizing Eq. (4.2) is not an option since $\boldsymbol{x} \in \mathcal{X}_{\text{OOD}}$ is unobserved. Thus, to efficiently solve Eq. (4.1) on $\mathcal{X}_{\text{OOD}}$, other strategies are required. We explore two such strategies: gradient-based search (Sec. 4.1) and generative replay (Sec. 4.2).

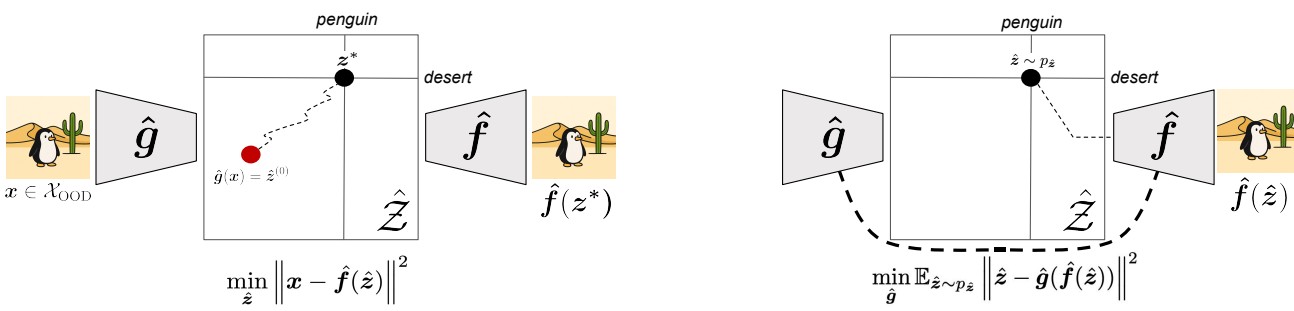

*Figure 4.* **Approaches for inverting a generator out-of-domain.** *Left.* Visualization of gradient-based search to invert a decoder $\hat{f}$ out-of-domain, with initialization given by an encoder $\hat{g}$. *Right.* Visualization of generative replay in which an encoder is trained on OOD images generated by a decoder.

### 4.1. Gradient-Based Search.

We note that the inference problem in Eq. (4.1) can be expressed as an optimization problem, i.e.,

$$z^* = \arg\min_{\hat{z}} \left\| x - \hat{f}(\hat{z}) \right\|^2. \qquad (4.3)$$

Thus, for OOD images $x \in \mathcal{X}_{\text{OOD}}$, we can recover $z^*$ *online* by solving Eq. (4.3) using *gradient-based optimization*. The efficiency of this, however, depends on the initialization $\hat{z}^{(0)}$. If $\hat{z}^{(0)}$ is far from the optimum, many gradient steps are required, leading to slow or suboptimal convergence. To mitigate this, we can leverage the encoder trained on $\mathcal{X}_{\text{ID}}$ to provide an initial prediction for $z^*$ such that $\hat{z}^{(0)} = \hat{g}(x)$ and then optimize Eq. (4.3) (see Fig. 4, left). Intuitively, the encoder gives a fast "System 1" guess that constrains the space for slower, "System 2" reasoning (Kahneman, 2011; Prabhudesai et al., 2023a), where "reasoning" corresponds to gradient-based search (LeCun, 2022).

### 4.2. Generative Replay

For out-of-domain images, Eq. (4.1) can also be solved in an *offline* manner by leveraging *generative replay* (Schwartenbeck et al., 2023; Wiedemer et al., 2024). Recall that images $x \in \mathcal{X}_{\text{OOD}}$ are generated by $f$ as combinations of ground-truth slots $z_k$. Since the decoder $\hat{f}$ identifies $f$ up to slot-wise transformations, images $x \in \mathcal{X}_{\text{OOD}}$ can likewise be generated by re-combining inferred slots $\hat{z}_k$. Concretely, this can be achieved by sampling a latent $\hat{z}$ from a distribution $p_{\hat{z}}$ with independent slot-wise marginals and decoding them with $\hat{f}$ such that $\hat{f}(\hat{z}) \in \mathcal{X}_{\text{OOD}}$. We can then solve Eq. (4.1) out-of-domain by training an encoder $\hat{g}$ on these samples such that $\hat{g}(\hat{f}(\hat{z})) = \hat{z}$ (see Fig. 4, right). This is captured by the following objective function (Wiedemer et al., 2024) (see Fig. 8 for pseudocode).

$$\min_{\hat{g}} \mathbb{E}_{\hat{z} \sim p_{\hat{z}}} \left\| \hat{z} - \hat{g}(\hat{f}(\hat{z})) \right\|^2. \qquad (4.4)$$

## 5. Experiments

In this section, we conduct an experimental study to assess (i) the extent to which compositional generalization emerges in non-generative methods without the necessary inductive biases § 5.1 and (ii) the extent to which generative methods, which leverage the inductive bias of a decoder, can achieve superior compositional generalization through gradient-based search and generative replay § 5.2.

**Datasets.** To evaluate compositional generalization, we require datasets that can be partitioned into ID and OOD regions containing unseen concept combinations. To this end, we leverage PUG (Bordes et al., 2023), which, to our knowledge, is the most visually complex dataset offering such controllability. Each image consists of a background and one or two animals, drawn from 10 and 32 categories, respectively. Using PUG, we construct three different ID/OOD datasets (see Fig. 7). In *PUG-Background*, OOD images contain unseen combinations of animals and backgrounds; in *PUG-Texture*, unseen combinations of animals and textures; and in *PUG-Object*, unseen combinations of animals.

**Encoders.** To map images to latent slots, we use encoders composed of two modules. First, images are divided into patches and mapped to a set of patch embeddings by a *base encoder*. Then, a *slot encoder* queries the resulting patch embeddings via cross-attention to produce a set of slots. We implement the base encoder using a Vision Transformer (ViT) (Dosovitskiy et al., 2021). For the slot encoder, we use either a cross-attention Transformer with self-attention between slots (Vaswani et al., 2017) or a Slot Attention module (Locatello et al., 2020). When using an unsupervised objective, however, we do not use Slot Attention as we found it to give inconsistent object disentanglement.

**Decoders.** To map latent slots to images, we use a cross-attention Transformer, where each pixel queries the slots to produce its output value (Sajjadi et al., 2022; van Steenkiste et al., 2024). Following Brady et al. (2025), we encourage this decoder to match Eq. (2.7) by regularizing the interactions between slots to be minimal. We achieve this by

*Table 1.* ID and OOD accuracy (%) for various non-generative methods across datasets. All models use a ViT-small base encoder.

| Slot Encoder | Objective | ID ↑ | OOD ↑ |
|---|---|---|---|
| *PUG-Background* | | | |
| Slot Attention | Supervised | 99.8 $_{\pm0.2}$ | 0.002 $_{\pm0.002}$ |
| Transformer | Supervised | 100.0 $_{\pm0.0}$ | 0.008 $_{\pm0.006}$ |
| Transformer | Unsupervised | 99.9 $_{\pm0.04}$ | 7.9 $_{\pm1.2}$ |
| *PUG-Texture* | | | |
| Slot Attention | Supervised | 99.9 $_{\pm0.042}$ | 53.2 $_{\pm6.4}$ |
| Transformer | Supervised | 100.0 $_{\pm0.0}$ | 64.8 $_{\pm6.6}$ |
| Transformer | Unsupervised | 100.0 $_{\pm0.0}$ | 56.3 $_{\pm2.7}$ |
| *PUG-Object* | | | |
| Slot Attention | Supervised | 100.0 $_{\pm0.0}$ | 98.0 $_{\pm1.0}$ |
| Transformer | Supervised | 100.0 $_{\pm0.0}$ | 95.9 $_{\pm2.0}$ |
| Transformer | Unsupervised | 99.9 $_{\pm0.1}$ | 99.9 $_{\pm0.04}$ |

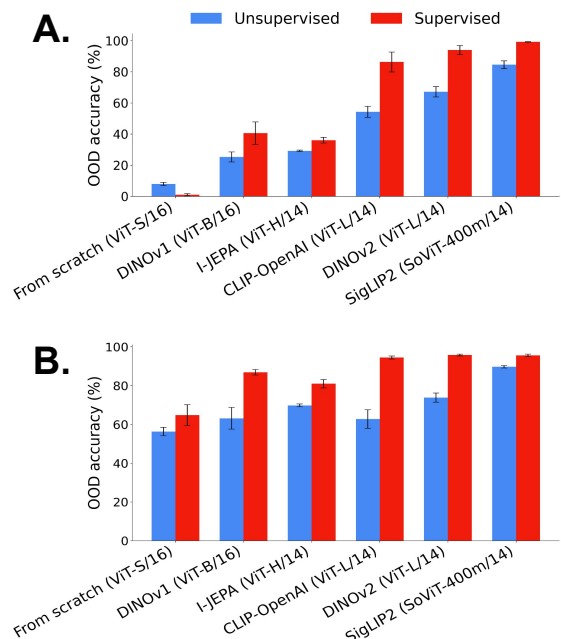

*Figure 5.* OOD performance on *PUG-Background* (**A.**) and *PUG-Texture* (**B.**) using different pretrained base encoders. Strong OOD performance emerges for non-generative methods on both datasets when using base encoders with large scale pretraining.

regularizing the attention weights of the model such that pixels are discouraged from attending to more than one slot (see § B.1 for details). In § C, we also report results using an unstructured decoder which is not designed to match $\mathcal{F}_{\text{int}}$.

**Metrics.** To measure whether a model's learned slots enable compositional generalization, we train a linear readout on ID slots to predict the animal and background labels associated with each ground-truth slot. A prediction is considered correct for a given image if all three labels are classified correctly. We use a single shared readout across slots and resolve slot-label assignments using the Hungarian algorithm (Kuhn, 1955). Compositional generalization is then quantified as the readout's accuracy on OOD images.

### 5.1. Non-Generative Methods

**Setup.** We evaluate the compositional generalization of non-generative methods trained on each dataset, using both supervised and unsupervised objectives. In the supervised setting, we train an encoder-only classifier to predict animal and background labels from slots. In the unsupervised setting, we jointly train encoders with the regularized Transformer decoder using a VAE loss (Kingma & Welling, 2014). Although encoders are trained to invert the decoder on ID data, the method remains non-generative OOD since the encoder is not designed to invert the decoder on OOD images.

**Results: Training from scratch.** In Tab. 1, we report ID and OOD accuracy across all datasets for each model and training objective. Across datasets, all methods achieve near-optimal ID accuracy. On both *PUG-Background* and *PUG-Texture*, however, OOD performance is poor, suggesting that further inductive biases are required for these methods to generalize compositionally. In contrast, on *PUG-Object*, we observe near-perfect OOD accuracy. In this dataset, OOD concept combinations do not interact, as animals never oc-

clude one another. This corresponds to the special case of $n = 0$ in § 3.1, where $\mathcal{G}_{\text{int}}$ has a simpler structure. Our results suggest that for such ground-truth structure, standard encoder inductive biases may be sufficient for OOD generalization.

**Results: Pretrained base encoder.** In Fig. 5, we study the impact of using base encoders with varying degrees of pretraining. We use DINOv1 (Caron et al., 2021), I-JEPA (Assran et al., 2023), DINOv2 (Oquab et al., 2024), CLIP (Radford et al., 2021), and SigLIP2 (Tschannen et al., 2025), which are optionally fine-tuned using a LoRA adapter (Hu et al., 2022). We report results for the best-performing combination of slot encoder and fine-tuning strategy. On *PUG-Background* (Fig. 5 A.), we observe OOD gains when using base encoders pretrained on relatively small corpora (e.g., DINOv1 or I-JEPA trained on ImageNet) compared to training from scratch. Performance continues to improve with larger-scale pretraining, as in DINOv2 and SigLIP2, as well as when encoders are trained with supervised data. A similar trend appears on PUG-Texture (Fig. 5 B.), though it is less pronounced. These results suggest that data scale can compensate for a lack of inductive biases in non-generative methods, enabling compositional generalization to emerge, but at the expense of data efficiency.

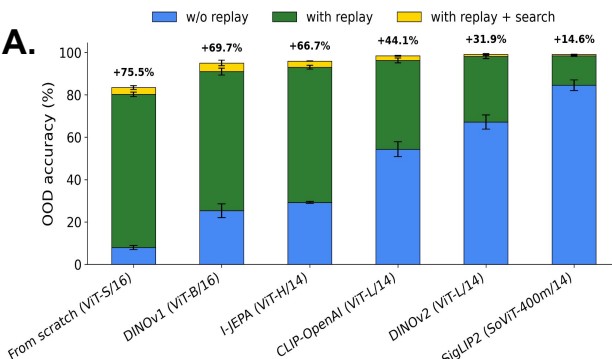 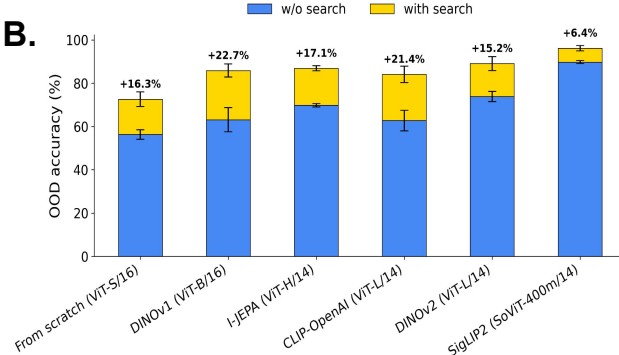

*Figure 6.* **OOD performance for generative methods.** We report OOD performance on PUG-Background and PUG-Texture for unsupervised VAEs which leverage replay (Sec. 4.2) and search (Sec. 4.1) trained with differing ViT base encoders. On PUG-Background (**A.**), we observe a significant increase in OOD performance using replay and additional gains through search. On PUG-Texture (**B.**) we also see a noticeable increase in OOD performance across all models when using search.

## 5.2. Generative Methods

**Setup.** We take the unsupervised VAEs trained in § 5.1 and apply search § 4.1 and replay § 4.2 to invert the decoder OOD. For replay, we generate OOD images by randomly sampling combinations of ID slots following Wiedemer et al. (2024) and optimize the encoder on these generations using Eq. (4.4). For search, we use the trained encoder to provide an initial representation which we then optimize using Eq. (4.3) with Adam (Kingma & Ba, 2015). For further details, see § B.1.

**Results.** On *PUG-Background* (Fig. 6 A.), we observe a significant increase in OOD accuracy when training encoders with replay across all base encoders, with further gains when search is additionally employed. On *PUG-Texture* (Fig. 6 B.), replay cannot be applied: in our setup, slots are designed to capture objects and backgrounds, and therefore cannot be trivially recomposed to generate novel animal–texture combinations. However, leveraging search yields a clear improvement in OOD performance across all models. These results highlight that generative methods can improve OOD performance by leveraging inductive biases, rather than relying solely on increased data scale. In Appendix C, we provide evidence that generative methods can achieve compositional generalization using less constrained Transformer decoders, highlighting that compositional generalization can also emerge from weaker, less explicit inductive biases.

## 6. Related Work

**Limitations of non-generative methods.** Several empirical studies have shown limitations in compositional generalization for non-generative methods trained using natural language supervision (Yuksekgonul et al., 2023; Lewis et al., 2024; Ma et al., 2023; Tong et al., 2024; Assouel et al., 2025;

West et al., 2024). These works generally posit that poor generalization arises from issues with standard contrastive language-image training objectives. In contrast, our theoretical and empirical contributions suggest that such issues are more fundamental, arising from the structure of the inverses of inverse generators $\mathcal{G}_{\text{int}}$.

**Generative approaches for improving generalization.** The idea that a generative approach can enable compositional generalization has long been advocated in the cognitive science community (Tenenbaum et al., 2011; Lake et al., 2017; 2015). Empirical realizations of this idea have recently been shown for diffusion models repurposed as classifiers (Wang et al., 2025; Jeong et al., 2025). Further (Prabhudesai et al., 2023a;b), showed that inverting a generative model with mechanisms similar to gradient-based search (Sec. 4.1) improves object-decomposition for OOD images and enhances the robustness of classifiers. Recent work explored training encoder-only models using synthetically generated data similar to Sec. 4.2, showing improvements in representations (Tian et al., 2023; Fan et al., 2025) and compositional generalization (Wiedemer et al., 2024; Assouel et al., 2022; Jung et al., 2024). Our work provides a theoretical motivation for these approaches by highlighting challenges in achieving compositional generalization using non-generative methods.

**Causal and anti-causal learning.** Our theoretical contribution relates to ideas in the field of causality. A key heuristic in this area posits that the factorization $P(\text{cause})P(\text{effect}|\text{cause})$ is, in general, less complex than the reverse factorization $P(\text{effect})P(\text{cause}|\text{effect})$ (Janzing & Schölkopf, 2010; Sun et al., 2006; 2008). It was conjectured by Kilbertus et al. (2018) that this principle indicates generalization is typically easier to achieve in the causal direction than in the anti-causal direction. Moreover, they propose an abstract version of the search procedure

(Sec. 4.1). The present paper can be seen as providing a formal justification for these ideas through theoretical insights on the structure of generators $f$ (the causal direction) and their inverses $g$ (the anti-causal direction).

## 7. Discussion

**Limitations.** Our theory is limited to generators which belong to $\mathcal{F}_{\text{int}}$. We studied this function class as it provides a suitable model of visual data and is the largest class which enables OOD identifiability. However, these results may, in principle, fail to generalize to function classes associated with other settings, where non-generative strategies may be effective. Additionally, while our experiments leverage photorealistic data, they focus on concepts in simple settings which do not fully capture the complexity of real world data. To this end, an important future question is to understand how to create benchmarks to evaluate compositional generalization in a rigorous manner on data at a more realistic scale. For further discussions, see § D.

**Conclusion.** In this work, we sought a principled understanding of whether compositional generalization should be pursued through generative or non-generative approaches. Theoretically, we showed that for non-generative methods, enforcing the structure needed to guarantee compositionality tends to be infeasible. As a result, generalization is determined largely by the optimization process rather than by principled guarantees. Empirically, we observed that methods optimized from scratch or with little pretraining data tend to fail at compositional generalization, while larger-scale pretrained models improve OOD performance at the cost of data efficiency. By contrast, generative approaches can directly enforce constraints for compositional generalization which manifest in significant gains in OOD performance in practice. While scaling such generative approaches to more challenging settings remains an open problem, we hope our findings will inspire renewed interest in this direction.

## Acknowledgments

We thank the anonymous reviewers, Julius von Kügelgen, Nicoló Zottino, Simon Schug, and Thaddäus Wiedmer for helpful feedback and discussions.

This work was supported by the German Federal Ministry of Education and Research (BMBF): Tübingen AI Center, FKZ: 01IS18039A, 01IS18039B. WB acknowledges financial support via an Emmy Noether Grant funded by the German Research Foundation (DFG) under grant no. BR 6382/1-1 and via the Open Philanthropy Foundation funded by the Good Ventures Foundation. WB is a member of the Machine Learning Cluster of Excellence, EXC number 2064/1 – Project number 390727645. This work utilized compute resources at the Tübingen Machine Learning Cloud, DFG FKZ INST 37/1057-1 FUGG.

## Impact Statement

This paper presents work whose goal is to advance the field of Machine Learning. There are many potential societal consequences of our work, none which we feel must be specifically highlighted here.

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

# Appendices

**Use of Language Models**    Large language models (LLMs) were employed exclusively during the final stages of manuscript preparation for the purpose of refining language, grammar, and readability. They were not used for generating ideas, conducting analysis, or contributing to the substantive content of this work.

## A. Proofs

In this section we collect the proofs of the results in the paper and some additional background material. First, in Section A.1 we investigate the local restrictions that $g \in \mathcal{G}_{\text{int}}$ need to satisfy. Similarly we investigate in Section A.2 whether we can enforce $g \in \mathcal{G}_{\text{int}}$ by architectural constraints. Let us, however, first introduce a notation for a subset of $\mathcal{F}_{\text{int}}$.

**Definition A.1** (Additive functions). We denote the function class of coordinate-wise additive functions $f : \mathbb{R}^{d_z} \to \mathbb{R}^{d_x}$ by $\mathcal{F}_{\text{add}}$. They can be expressed as

$$f(x) = \sum_{i=1}^{d_z} f_i(x_i) \tag{A.1}$$

where $f_i : \mathbb{R} \to \mathbb{R}^{d_x}$.

Clearly $\mathcal{F}_{\text{add}}$ agrees with $\mathcal{F}_{\text{int}}$ for $n = m = 1$, i.e., interactions of first order and blocks of dimension 1 and generally $\mathcal{F}_{\text{add}} \subset \mathcal{F}_{\text{int}}$ for $n \geq 1$ (higher order interactions and larger blocks are more flexible).

### A.1. Structure of $\mathcal{G}_{\text{int}}$

As discussed in the main text, we can enforce $f \in \mathcal{F}_{\text{int}}$ by enforcing that certain derivatives of $f$ vanish (see equation 3.2). We now study to what extend this generalizes to functions $g \in \mathcal{G}_{\text{int}}$ that are left inverses of such functions.

**The key relation.**    The key relation that we need for the proofs below is that if $g \circ f(z) = z$ for two functions $f : \mathbb{R}^{d_z} \to \mathbb{R}^{d_x}$ and $g : \mathbb{R}^{d_x} \to \mathbb{R}^{d_z}$, then for every $s \in [d_z]$

$$Df^\top(z)D^2 g_s(f(z))Df(z) + \sum_{k=1}^{d_x}(\partial_k g_s)(f(z))D^2 f_k(z) = 0. \tag{A.2}$$

This relation follows by straightforward calculation, indeed we find using the chain rule

$$
\begin{aligned}
\partial_i \partial_j g_s(f(z)) &= \partial_i \left( \sum_{k=1}^{d_x} \partial_j f_k(z)(\partial_k g_s)(f(z)) \right) \\
&= \sum_{k,l=1}^{d_x} \partial_j f_k(z)\partial_i f_l(z)(\partial_k \partial_l g_s)(f(z)) + \sum_{k=1}^{d_x} \partial_i \partial_j f_k(z)(\partial_k g_s)(f(z))
\end{aligned}
\tag{A.3}
$$

which is equation A.2 after rewriting the relation in matrix form.

**Restrictions for $d_x = d_z$.**    We now prove Lemma 3.1 showing that for $d_x = d_z$, i.e., for equal dimension of latent space and data it is possible to find a local constraint for the inverses of additive functions $f \in \mathcal{F}_{\text{add}}$.

**Lemma A.2.** *Let $g \in \mathcal{G}_{\text{int}}$ for $n = m = 1$ and $d_x = d_z$. Then $g$ has the property that for $x \in \mathcal{X}$*

$$(Dg)^{-\top}(x)D^2 g_s(x)(Dg)^{-1}(x) \in \text{Diag}(d_x) \tag{3.3}$$

*is a diagonal matrix for $s \in [d_z]$. Further, if $g$ is a diffeomorphism satisfying Eq. (3.3) then $g \in \mathcal{G}_{\text{int}}$.*

*Remark* A.3. For higher dimensional slots there is a natural generalization, namely, the expression $D\boldsymbol{g}^{-\top}D^2\boldsymbol{g}_s D\boldsymbol{g}^{-1}$ has a block diagonal structure.

*Proof.* Note that $\boldsymbol{g}\circ\boldsymbol{f}(\boldsymbol{z})=\boldsymbol{z}$ implies $\mathrm{Id}_{d_z}=(D\boldsymbol{g}\circ\boldsymbol{f})D\boldsymbol{f}$ and thus $D\boldsymbol{f}(z)=(D\boldsymbol{g})^{-1}(\boldsymbol{f}(\boldsymbol{z}))$. Therefore, we find using equation A.2

$$(D\boldsymbol{g})^{-\top}(\boldsymbol{f}(\boldsymbol{z}))D^2\boldsymbol{g}_s(\boldsymbol{f}(\boldsymbol{z}))(D\boldsymbol{g})^{-1}(\boldsymbol{f}(\boldsymbol{z}))=-\sum_{k=1}^{d_x}(\partial_k\boldsymbol{g}_s)(\boldsymbol{f}(\boldsymbol{z}))D^2\boldsymbol{f}_k(\boldsymbol{z})\in\mathrm{Diag}(d_z). \tag{A.4}$$

where we used that $\boldsymbol{f}\in\mathcal{F}_{\mathrm{add}}$ implies that the off-diagonal entries of $D^2\boldsymbol{f}$ vanish. This implies the first part of the statement. For the reverse statement, we apply equation A.2 to $\boldsymbol{f}\circ\boldsymbol{g}(\boldsymbol{x})=\boldsymbol{x}$ (here we use $d_z=d_x$) and we find that

$$0=(D\boldsymbol{g})^{\top}(\boldsymbol{x})D^2\boldsymbol{f}_s(\boldsymbol{g}(\boldsymbol{x}))D\boldsymbol{g}(\boldsymbol{x})+\sum_{k=1}^{d_z}(\partial_k\boldsymbol{f}_s)(\boldsymbol{g}(\boldsymbol{x}))D^2\boldsymbol{g}_k(\boldsymbol{x}). \tag{A.5}$$

We multiply this relation from the left and right by $(D\boldsymbol{g})^{-\top}(\boldsymbol{x})$ and $(D\boldsymbol{g})^{-1}(\boldsymbol{x})$ respectively (the inverses exist by assumption) and we find

$$D^2\boldsymbol{f}_s(\boldsymbol{g}(\boldsymbol{x}))=-\sum_{k=1}^{d_z}(\partial_k\boldsymbol{f}_s)(\boldsymbol{g}(\boldsymbol{x}))(D\boldsymbol{g})^{-\top}(\boldsymbol{x})D^2\boldsymbol{g}_k(\boldsymbol{x})(D\boldsymbol{g})^{-1}(\boldsymbol{x})\in\mathrm{Diag}(d_z). \tag{A.6}$$

Here we used the assumption equation 3.3 to conclude that the right hand side is diagonal. Therefore $\boldsymbol{f}$ has a diagonal Hessian which implies that it is additive. $\square$

The previous statement can be generalized to the general case $d_x>d_z$. The crucial ingredient is the following simple and standard lemma.

**Lemma A.4.** *Let* $A\in\mathbb{R}^{d_1\times d_2}$ *and* $B\in\mathbb{R}^{d_2\times d_1}$ *two matrices with* $d_2\geq d_1$ *and assume that* $AB=\mathbf{1}_{d_1\times d_1}$. *Then* $B=(A\Pi)^+$ *where* $\Pi$ *denotes the orthogonal projection onto* $\mathrm{Range}(B)$ *and* $(\cdot)^+$ *the Moore-Penrose inverse of a matrix.*

*Proof.* We check that $B$ satisfies the Moore-Penrose axioms ($MM^+M=M$, $M^+MM^+=M^+$, $M^+M$ and $MM^+$ are Hermitian). We find

$$A\Pi BA\Pi=ABA\Pi=A\Pi \tag{A.7}$$

where we used $\Pi B=B$ by definition of $\Pi$. Similarly, we obtain

$$BA\Pi B=\Pi B=B. \tag{A.8}$$

Next we claim that

$$BA\Pi=\Pi \tag{A.9}$$

which is Hermitian. Consider $v\in\mathbb{R}^{d_2}$ then by definition of $\Pi$ there is $w\in\mathbb{R}^{d_1}$ such that $Bw=\Pi v$ and thus

$$BA\Pi v=BABw=Bw=\Pi v. \tag{A.10}$$

Finally, we find

$$A\Pi B=AB=\mathbf{1}_{d_1\times d_1}. \tag{A.11}$$

$\square$

We have the following generalization of Lemma 3.1.

**Lemma A.5.** *Let $\boldsymbol{f} \in \mathcal{F}_{\text{add}}$ and $\boldsymbol{g}$ a left-inverse of $\boldsymbol{f}$. Then $\boldsymbol{g}$ has the property that for $\boldsymbol{x} \in \mathcal{X}$*

$$\left( (D\boldsymbol{g}(\boldsymbol{x})\Pi_{T_{\boldsymbol{x}}\mathcal{X}})^+ \right)^\top D^2\boldsymbol{g}_s(\boldsymbol{x}) \, (D\boldsymbol{g}(\boldsymbol{x})\Pi_{T_{\boldsymbol{x}}\mathcal{X}})^+ \in \text{Diag}(d_z) \tag{A.12}$$

*is a diagonal matrix for $s \in [d_z]$. Here, we denote by $\Pi_{T_{\boldsymbol{x}}\mathcal{X}}$ the orthogonal projection on the tangent space at $\boldsymbol{x}$.*

*Proof.* Starting from equation A.2 we find that for $\boldsymbol{f} \in \mathcal{F}_{\text{add}}$ we get

$$D\boldsymbol{f}^\top(\boldsymbol{z})D^2\boldsymbol{g}_s(\boldsymbol{f}(\boldsymbol{z}))D\boldsymbol{f}(\boldsymbol{z}) \in \text{Diag}(d_z). \tag{A.13}$$

Applying Lemma A.4 we find

$$D\boldsymbol{f}(\boldsymbol{z}) = \left( D\boldsymbol{g}(\boldsymbol{f}(\boldsymbol{z}))\Pi_{T_{\boldsymbol{f}(\boldsymbol{z})}\mathcal{X}} \right)^+ \tag{A.14}$$

because the range of $D\boldsymbol{f}$ is the tangent space of the data manifold. Therefore we conclude that for $\boldsymbol{x} \in \mathcal{X}$ the relation equation A.12 indeed holds. $\square$

**Regularization for $d_x > d_z$.** In this paragraph we investigate the local restrictions that $\boldsymbol{g} \in \mathcal{G}_{\text{int}}$ need to satisfy, and in particular we prove Theorem 3.2. The proof of Theorem 3.2 requires two lemmas as a key ingredient, which state that the crucial constraint on the second derivative stated in equation A.2 can be satisfied for a suitable choice of $\boldsymbol{M} = D\boldsymbol{f}(0)$ and $D^2\boldsymbol{f}(0)$ for given matrices $\boldsymbol{B}_s$ corresponding to the Hessian of $\boldsymbol{g}$ and almost every matrix $\boldsymbol{A}$ (corresponding to the Jacobian of $\boldsymbol{g}$). The first lemma establishes the existence of $\boldsymbol{M}$ such that first term in equation A.2 (given by $\boldsymbol{M}^\top \boldsymbol{B}_s \boldsymbol{M}$ is diagonal for all $s$. The second lemma constructs suitable second derivatives $D^2\boldsymbol{f}$ so that the relation equation A.2 also holds for the diagonal entries.

**Lemma A.6.** *Assume $d_x \geq d_z^3$. For all symmetric matrices $\boldsymbol{B}_s \in \mathbb{R}^{d_x \times d_x}$ for $s \in [d_z]$, and almost every $\boldsymbol{A} \in \mathbb{R}^{d_z \times d_x}$ there is a matrix $\boldsymbol{M} \in \mathbb{R}^{d_x \times d_z}$ such that $\boldsymbol{M}^\top \boldsymbol{B}_s \boldsymbol{M} \in \text{Diag}(d_z)$ for $s \in [d_z]$ and $\boldsymbol{A}\boldsymbol{M} = \text{Id}_{d_z}$.*

*Remark* A.7. 1. Counting parameters and equations, we find that $\boldsymbol{M}$ has $d_z d_x$ parameters and (by symmetry of $\boldsymbol{B}_s$) there are

$$d_z \cdot \frac{d_z(d_z - 1)}{2} + d_z^2 = \frac{d_z^2(d_z + 1)}{2} \tag{A.15}$$

equations. So, generally, we expect the result to hold for $d_x \geq d_z(d_z + 1)/2$.

2. On the other hand, the result does not hold for every $\boldsymbol{A}$ with maximal rank. Indeed, there can be a non-trivial null set of full rank matrices $\boldsymbol{A}$ such that the result does not hold. E.g., consider $d_z = 2$, $\boldsymbol{A} \in \mathbb{R}^{d_z \times d_x}$ such that all entries of $\boldsymbol{A}$ are zero except $\boldsymbol{A}_{1,1} = \boldsymbol{A}_{2,2} = 1$. Moreover, $\boldsymbol{B}_1$ has all entries zero except $(\boldsymbol{B}_1)_{1,2} = (\boldsymbol{B}_1)_{2,1} = 1$. Then $\boldsymbol{A}\boldsymbol{M} = \text{Id}_{d_z}$ implies that $\boldsymbol{M}_{1,1} = \boldsymbol{M}_{2,2} = 1$, and $\boldsymbol{M}_{1,2} = \boldsymbol{M}_{2,1} = 0$. But then we find $\boldsymbol{M}_{:,1}^\top \boldsymbol{B}_1 \boldsymbol{M}_{:,2} = (\boldsymbol{B}_1)_{1,2} = 1 \neq 0$.

*Proof.* We inductively construct $d_z$ linear subspaces $V_i \subset \mathbb{R}^{d_x}$ such that $\dim(V_i) = d_z$ and

$$(\boldsymbol{v}^i)^\top \boldsymbol{B}_s \boldsymbol{v}^j = 0 \tag{A.16}$$

for $\boldsymbol{v}^i \in V_i$, $\boldsymbol{v}^j \in V_j$ and $i \neq j$. We pick $V_1$ arbitrarily. Then, given a basis $\boldsymbol{v}^{i,1}, \ldots, \boldsymbol{v}^{i,d_z}$ of $V_i$ for $i \leq j$ we select $V_{j+1} \subset \ker \boldsymbol{T}_j$ where $\boldsymbol{T}_j : \mathbb{R}^{d_x} \to \mathbb{R}^{d_z^2 \cdot j}$ given by $(\boldsymbol{T}_j \boldsymbol{v})_{s,(k,i)} = (\boldsymbol{v}^{i,k})^\top \boldsymbol{B}_s \boldsymbol{v}$ (here it is convenient to identify $[d_z^2 \cdot j]$ with $[d_z] \times ([d_z] \times [j])$). By assumption $d_x - d_z^2 \cdot j \geq d_x - d_z^2 \cdot (d_z - 1) \geq d_z$ and therefore $\dim \ker \boldsymbol{T}_j \geq d_z$ and we can find a suitable subspace $V_{j+1} \subset \ker \boldsymbol{T}_j$. Given a matrix $\boldsymbol{A} = (\boldsymbol{a}^1, \ldots, \boldsymbol{a}^{d_z})^\top \in \mathbb{R}^{d_z \times d_x}$, we want to find $\boldsymbol{w}^i \in V_i$ so that $\boldsymbol{M} = (\boldsymbol{w}^1, \ldots, \boldsymbol{w}^{d_z})$ satisfies $\boldsymbol{A}\boldsymbol{M} = \text{Id}_{d_z}$. Equivalently $\boldsymbol{A}\boldsymbol{w}^i = \boldsymbol{e}^i$, where $\boldsymbol{e}^i$ denotes the $i$-th standard basis vector. We expand into the basis of $V_i$, i.e., $\boldsymbol{w}^i = \sum_j \boldsymbol{\lambda}_j^i \boldsymbol{v}^{i,j}$ and find the equivalent relation

$$\boldsymbol{A}\boldsymbol{w}^i = (\boldsymbol{a}^1, \ldots, \boldsymbol{a}^{d_z})^\top (\boldsymbol{v}^{i,1}, \ldots, \boldsymbol{v}^{i,d_z})\boldsymbol{\lambda}^i = \boldsymbol{e}^i. \tag{A.17}$$

Since the second matrix has maximal rank ($(\boldsymbol{v}^{i,k})_{1 \leq k \leq d_z}$ is a basis of $V_i$), we find that for almost all $\boldsymbol{A}$ the matrix product is invertible, and a solution $\boldsymbol{\lambda}^i$ exists and thus a suitable $\boldsymbol{w}^i$ exists. To see this, we can assume that the basis $\boldsymbol{v}^{i,\cdot}$ is an orthonormal basis and expand $\boldsymbol{a}_i$ in this basis (and an irrelevant orthogonal complement). We conclude that for almost all $\boldsymbol{A}$ such a $\boldsymbol{w}^i$ exists. Since the union of null-sets is a null-set the same statement holds for almost all $\boldsymbol{A}$ for all $i$ at the same time and therefore we find a matrix $\boldsymbol{M}$ such that $\boldsymbol{A}\boldsymbol{M} = \text{Id}_{d_z}$ and, moreover, $(\boldsymbol{w}^i)^\top \boldsymbol{B}_s \boldsymbol{w}^j = 0$ because this holds for all $\boldsymbol{w}^i \in V_i$ and $\boldsymbol{w}^j \in V_j$. $\square$

We now construct the diagonal matrices that will later correspond to $D^2 \boldsymbol{f}_s$.

**Lemma A.8.** *Assume $d_x \geq d_z$ Given $\boldsymbol{A} \in \mathbb{R}^{d_z \times d_x}$ of maximal rank and diagonal matrices $\boldsymbol{D}^1, \ldots, \boldsymbol{D}^{d_z} \in \mathbb{R}^{d_z \times d_z}$ we can find diagonal matrices $\boldsymbol{\Lambda}^1, \ldots, \boldsymbol{\Lambda}^{d_x} \in \mathbb{R}^{d_z \times d_z}$ such that for all $s \in [d_z]$*

$$\boldsymbol{D}^s = -\sum_{i=1}^{d_x} \boldsymbol{A}_{s,i} \boldsymbol{\Lambda}^i. \tag{A.18}$$

*Proof.* The proof is straightforward as soon as one observes that this is a linear equation for the diagonal entries of $\boldsymbol{\Lambda}^i$. Indeed, denoting by $\boldsymbol{\lambda} = (\boldsymbol{\Lambda}_{11}^1, \ldots, \Lambda_{d_z,d_z}^1, \ldots, (\Lambda_{d_z,d_z}^{d_x})^\top \in \mathbb{R}^{d_z \cdot d_x}$ the vector containing all diagonal entries of the matrices $\boldsymbol{\Lambda}^i$ and similarly $\boldsymbol{d} = (\boldsymbol{D}_{11}^1, \ldots, \boldsymbol{D}_{d_z,d_z}^1, \ldots, \boldsymbol{D}_{d_z,d_z}^{d_z})^\top \in \mathbb{R}^{d_z^2}$ for the diagonal entries of $\boldsymbol{D}^s$. Then we can rewrite equation A.18 as follows using the Kronecker product $\otimes$

$$(\boldsymbol{A} \otimes \mathrm{Id}_{d_z}) \boldsymbol{\lambda} = -\boldsymbol{d}. \tag{A.19}$$

Now the rank of the matrix $\boldsymbol{A} \otimes \mathrm{Id}_{d_z}$ is the product of the ranks, i.e., $d_z \min(d_x, d_z) = d_z^2 \leq d_x d_z$ and thus a solution $\boldsymbol{\lambda}$ exists. $\qquad\square$

With these technical lemmas at hand, we can prove the theorem which we now restate for convenience of the reader.

**Theorem 3.2.** *Assume that $d_x \geq d_z^3$. Let $\boldsymbol{B}_l \in \mathbb{R}^{d_x \times d_x}$ be symmetric matrices for $1 \leq l \leq d_z$. Then there is for any $\boldsymbol{x}_0 \in \mathbb{R}^{d_x}$ and for almost every $\boldsymbol{A} \in \mathbb{R}^{d_z \times d_x}$ a generator $\boldsymbol{f} \in \mathcal{F}_{\mathrm{int}}$ with a (left)-inverse $\boldsymbol{g} \in \mathcal{G}_{\mathrm{int}}$, such that $\boldsymbol{f}(0) = \boldsymbol{x}_0$ and $D\boldsymbol{g}(\boldsymbol{x}_0) = \boldsymbol{A}$ and $D^2 \boldsymbol{g}_l(\boldsymbol{x}_0) = \boldsymbol{B}_l$ for $1 \leq l \leq d_z$.*

*Proof of Theorem 3.2.* Clearly we can assume that $\boldsymbol{x}_0 = 0$. The key idea is that if we can ensure that equation A.2 holds for $\boldsymbol{z} = 0$ we can extend $\boldsymbol{f}$ and $\boldsymbol{g}$ such that $\boldsymbol{g} \circ \boldsymbol{f}(\boldsymbol{z}) = \boldsymbol{z}$ and $\boldsymbol{f} \in \mathcal{F}_{\mathrm{add}}$. To achieve this, we first apply Lemma A.6 and then find a matrix $\boldsymbol{M}$ such that $\boldsymbol{A}\boldsymbol{M} = \mathrm{Id}_{d_z}$ and $\boldsymbol{M}^\top \boldsymbol{B}_s \boldsymbol{M} \in \mathrm{Diag}(d_z)$. Then we apply Lemma A.8 and find matrices $\boldsymbol{\Lambda}^i$ such that

$$\boldsymbol{M}^\top \boldsymbol{B}_s \boldsymbol{M} + \sum_{i=1}^{d_x} \boldsymbol{A}_{s,i} \boldsymbol{\Lambda}^i = 0. \tag{A.20}$$

Now we pick a function $\boldsymbol{f} \in \mathcal{F}_{\mathrm{add}}$ such that $\boldsymbol{f}(0) = 0$, $D\boldsymbol{f}(0) = \boldsymbol{M}$ and $D^2 \boldsymbol{f}_i = \boldsymbol{\Lambda}^i$. Clearly, this is possible because $\boldsymbol{\Lambda}^i$ are diagonal, e.g., we can locally use a quadratic polynomial to achieve this. The next step is to construct a function $\boldsymbol{g}$ such that $\boldsymbol{g} \circ \boldsymbol{f}(\boldsymbol{z}) = \boldsymbol{z}$. Using standard techniques (partition of unity) it is sufficient to construct this locally and then extend it globally. First, we consider $\bar{\phi} : \Omega \subset \mathbb{R}^{d_x} \times \mathbb{R}^{d_x}$ such that $\bar{\phi}(\boldsymbol{f}(\boldsymbol{z})) = (\boldsymbol{z}, 0)$ for $\boldsymbol{z} \in \Omega$ (e.g., by the existence of tubular neighborhoods). We call the composition of $\bar{\phi}$ with the projection on the first $d_z$ components $\phi$. Then we define

$$\boldsymbol{g}(\boldsymbol{x}) = \boldsymbol{A}\boldsymbol{x} + \frac{1}{2} \begin{pmatrix} \boldsymbol{x}^\top \boldsymbol{B}_1 \boldsymbol{x} \\ \vdots \\ \boldsymbol{x}^\top \boldsymbol{B}_{d_z} \boldsymbol{x} \end{pmatrix} + \boldsymbol{h}(\phi(\boldsymbol{x})) \tag{A.21}$$

where $\boldsymbol{h}$ is given by

$$\boldsymbol{h}(\boldsymbol{z}) = \boldsymbol{h}(\phi(\boldsymbol{f}(\boldsymbol{z})) = \boldsymbol{z} - \boldsymbol{A}\boldsymbol{f}(\boldsymbol{z}) - \frac{1}{2} \begin{pmatrix} \boldsymbol{f}(\boldsymbol{z})^\top \boldsymbol{B}_1 \boldsymbol{f}(\boldsymbol{z}) \\ \vdots \\ \boldsymbol{f}(\boldsymbol{z})^\top \boldsymbol{B}_{d_z} \boldsymbol{f}(\boldsymbol{z}). \end{pmatrix} \tag{A.22}$$

We can calculate

$$\boldsymbol{g}(\boldsymbol{f}(\boldsymbol{z})) = \boldsymbol{z} \tag{A.23}$$

so $\boldsymbol{g}$ is indeed a left-inverse of $\boldsymbol{f}$. Taking the derivative of this equation at 0 we obtain

$$D\boldsymbol{h}(0) = \mathrm{Id}_{d_z} - \boldsymbol{A}(D\boldsymbol{f}(0)) + 0 = \mathrm{Id}_{d_z} - \boldsymbol{A}\boldsymbol{M} = 0. \tag{A.24}$$

For the second derivative we get

$$D^2 \boldsymbol{h}_s(0) = - \sum_{i=1}^{d_x} \boldsymbol{A}_{s,i} D^2 \boldsymbol{f}_i(\boldsymbol{z}) - D\boldsymbol{f}(0)^\top \boldsymbol{B}_s D\boldsymbol{f}(0) = - \sum_{i=1}^{d_x} \boldsymbol{A}_{s,i} \boldsymbol{\Lambda}^i + \boldsymbol{M}^\top \boldsymbol{B}_s \boldsymbol{M} = 0. \tag{A.25}$$

Here we used for the derivative of the quadratic term that the contribution where the derivative hits one $\boldsymbol{f}(\boldsymbol{z})$ twice vanishes since $\boldsymbol{f}(0) = 0$. Finally, we can now evaluate

$$D\boldsymbol{g}(0) = \boldsymbol{A} + D\boldsymbol{h}(\phi(0)) = \boldsymbol{A} + D\boldsymbol{h}(0) = \boldsymbol{A} \tag{A.26}$$

and

$$D^2 \boldsymbol{g}_s(0) = \boldsymbol{B}_s + D^2 \boldsymbol{h}_s \circ \phi = \boldsymbol{B}_s \tag{A.27}$$

where $D^2 \boldsymbol{h}_s \circ \phi = 0$ follows from the chain rule and $D\boldsymbol{h}(0) = 0$ and $D^2 \boldsymbol{h}(0) = 0$. □

## A.2. Constraining $\mathcal{G}_{\text{enc}}$ by architecture

In this section we discuss results showing that it is challenging to construct practical function classes $\mathcal{G}_{\text{enc}}$ which are sufficiently expressive so that they contain a left-inverse for each $\boldsymbol{f} \in \mathcal{F}_{\text{int}}$. As explained before, the main challenge is that setting $\mathcal{G}_{\text{enc}} = \mathcal{G}_{\text{int}}$ is in principle sufficient to ensure identifiability and out of distribution generalization. So we need to make additional assumptions on $\mathcal{G}_{\text{enc}}$ that function classes used in widely applied algorithms satisfy which then ensure that $\mathcal{G}_{\text{enc}}$ is very expressive preventing that equation 2.6 holds. We will make the assumption that $\mathcal{G}_{\text{enc}}$ has a linear structure, i.e., $\boldsymbol{g}_1 + \boldsymbol{g}_2 \in \mathcal{G}_{\text{enc}}$ if $\boldsymbol{g}_1, \boldsymbol{g}_2 \in \mathcal{G}_{\text{enc}}$. This is clearly satisfied when $\mathcal{G}_{\text{enc}}$ is a vector space (e.g., this assumption is satisfied for linear or kernel methods, or when learning a linear head on a fixed representation). For functions implemented by neural networks with fixed architecture this is in general not true. However, it does apply to infinite width limits of fixed architectures (this does not generally imply universal approximation properties when the architecture is sparse, e.g., we use slot-wise neural networks for the forward direction which cannot approximate interactions $\boldsymbol{x}_1 \boldsymbol{x}_2$ even at infinite width). Note that large width is also generally required to make neural networks sufficiently expressive because for fixed width neural networks implement a parametric function class while $\mathcal{F}_{\text{int}}$ is non-parametric. We then show that such a function class $\mathcal{G}_{\text{enc}}$ does not have a useful inductive bias.

**Architecture constraints for $d_x = d_z$.** We first consider the simpler case $d_x = d_z$ where $\boldsymbol{f}$ is bijective on the codomain (and not only on its image).

Our main result here is that $\mathcal{G}_{\text{enc}}$ has the universal approximation property when $\mathcal{G}_{\text{enc}} \supset \mathcal{F}_{\text{add}}^{-1}$ and $\mathcal{G}_{\text{enc}}$ is closed under addition.

**Theorem A.9.** *Assume $d_x = d_z = d$. Consider an encoder function class $\mathcal{G}_{\text{enc}}$ with the following two properties:*

1. *The class $\mathcal{G}_{\text{enc}}$ is closed under addition, i.e., for $\boldsymbol{g}_1, \boldsymbol{g}_2 \in \mathcal{G}_{\text{enc}}$ also $\boldsymbol{g}_1 + \boldsymbol{g}_2 \in \mathcal{G}_{\text{enc}}$.*

2. *The function class $\mathcal{G}_{\text{enc}}$ is expressive enough such that it contains all inverses of additive functions, i.e., $\mathcal{F}_{\text{add}}^{-1} \subset \mathcal{G}_{\text{enc}}$.*

*Then $\mathcal{G}_{\text{enc}}$ is dense in the space of all continuous functions on all compact subset of $\mathbb{R}^{d_x}$.*

Since $\mathcal{F}_{\text{add}} \subset \mathcal{F}_{\text{int}}$ for $n \geq 1$ and any $m$ we directly get the following corollary.

**Corollary A.10.** *Assume $d_x = d_z = d$ and the encoder function class $\mathcal{G}_{\text{enc}}$ is closed under addition and satisfies $\mathcal{G}_{\text{int}} \subset \mathcal{G}_{\text{enc}}$. Then $\mathcal{G}_{\text{enc}}$ is dense in the space of all continuous functions on all compact subset of $\mathbb{R}^{d_x}$.*

The takeaway from these results is that it is challenging to find natural function classes $\mathcal{G}_{\text{enc}}$ so that $\mathcal{G}_{\text{enc}} \supset \mathcal{G}_{\text{int}}$ (sufficient expressivity) but $\mathcal{G}_{\text{enc}}$ is not much larger than $\mathcal{G}_{\text{int}}$. Therefore, learning only encoders from $\mathcal{G}_{\text{enc}}$ does not provide a strong inductive bias towards the inverse of the ground truth decoder and out of distribution generalization.

*Proof of Theorem A.9.* The general strategy is to prove that the conditions imply that all maps $\boldsymbol{g}$ where $\boldsymbol{g}_j$ (the $j$-th coordinate of $\boldsymbol{g}$) is any polynomial and $\boldsymbol{g}_i = 0$ for $i \neq j$ are contained in $\mathcal{G}_{\text{enc}}$. This will end the proof because polynomials are dense in the scalar valued continuous functions, and we can then apply this result coordinate-wise using the additive structure.

**Step 1:** Vector space structure of $\mathcal{G}_{\text{enc}}$. We now show that we can scale certain functions in $\mathcal{G}_{\text{enc}}$. Denote for $\lambda \neq 0$ by $M_\lambda$ the multiplication map $z \to \lambda z$. Then $\boldsymbol{f} \circ M_\lambda \in \mathcal{F}_{\text{add}}$ if $\boldsymbol{f} \in \mathcal{F}_{\text{add}}$. Since $(\boldsymbol{f} \circ M_\lambda)^{-1} = \lambda^{-1}\boldsymbol{f}^{-1}$ we conclude that scalar multiples of $\boldsymbol{f}^{-1}$ are in $\mathcal{F}_{\text{add}}^{-1}$ and the first assumption then implies that the vector space $V$ generated by $\boldsymbol{f}^{-1}$ for $\boldsymbol{f} \in \mathcal{F}_{\text{add}}$ is contained in $\mathcal{G}_{\text{enc}}$.

**Step 2:** We show that the monomials $x_i^k$ are contained in $\mathcal{G}_{\text{enc}}$. Consider the map $\boldsymbol{f} \in \mathcal{F}_{\text{add}}$ where $\boldsymbol{x} = \boldsymbol{f}(\boldsymbol{z})$ has coordinates

$$\begin{aligned}
\boldsymbol{x}_1 &= \boldsymbol{z}_2^k + \boldsymbol{z}_1 \\
\boldsymbol{x}_i &= \boldsymbol{z}_i \quad \text{for } d \geq i \geq 2.
\end{aligned} \tag{A.28}$$

This is clearly an additive function with inverse

$$\begin{aligned}
\boldsymbol{z}_1 &= \boldsymbol{x}_1 - \boldsymbol{x}_2^k \\
\boldsymbol{z}_i &= \boldsymbol{x}_i \quad \text{for } d \geq i \geq 2.
\end{aligned} \tag{A.29}$$

Similarly, we consider

$$\begin{aligned}
\boldsymbol{x}_1 &= -(-\boldsymbol{z}_2)^k - \boldsymbol{z}_1 \\
\boldsymbol{x}_i &= -\boldsymbol{z}_i \quad \text{for } d \geq i \geq 2.
\end{aligned} \tag{A.30}$$

with inverse

$$\begin{aligned}
\boldsymbol{z}_1 &= -\boldsymbol{x}_1 - \boldsymbol{x}_2^k \\
\boldsymbol{z}_i &= -\boldsymbol{x}_i \quad \text{for } d \geq i \geq 2.
\end{aligned} \tag{A.31}$$

Summing these two functions, we find that the function $\boldsymbol{g}$ with

$$\boldsymbol{g}_i(x) = -2\delta_{1i}\boldsymbol{x}_1^k \tag{A.32}$$

satisfies $\boldsymbol{g} \in \mathcal{G}_{\text{enc}}$. By permutation of the outputs and inputs (and scaling) we find that all functions $\boldsymbol{g}$ with $\boldsymbol{g}_i(\boldsymbol{x}) = \delta_{ij}\boldsymbol{x}_l^k$ are in $\mathcal{G}_{\text{enc}}$ for all $j, l \in [d]$ and $k \in \mathbb{N}$.

**Step 3:** Now we show with a similar argument that more generally functions of the form $\boldsymbol{g}_j(\boldsymbol{x}) = \delta_{jl}(\sum_{i=1}^d \boldsymbol{\alpha}_i \boldsymbol{x}_i)^k$ for all coefficients $\boldsymbol{\alpha}_i$ and all $1 \leq l \leq d$ are in $\mathcal{G}_{\text{enc}}$. If only one $\boldsymbol{\alpha}_i$ is non-zero we have shown this before, so we can assume that at least two $\boldsymbol{\alpha}_i$ are non-zero and without loss of generality we assume that $\boldsymbol{\alpha}_i$ for $1 \leq i \leq k$ are non-zero where $2 \leq k \leq d$. Then we consider the additive map $\boldsymbol{g}$ which satisfies for $\boldsymbol{x} = \boldsymbol{g}(\boldsymbol{z})$

$$\begin{aligned}
\boldsymbol{x}_1 &= \frac{1}{\boldsymbol{\alpha}_1}\left(\boldsymbol{z}_1^k + \boldsymbol{z}_1 - \sum_{i=2}^k \boldsymbol{z}_i\right), \\
\boldsymbol{x}_2 &= \frac{1}{\boldsymbol{\alpha}_2}(\boldsymbol{z}_2 - \boldsymbol{z}_1^k), \\
\boldsymbol{x}_i &= \frac{1}{\boldsymbol{\alpha}_i}\boldsymbol{z}_i \quad \text{for } 3 \leq i \leq k, \\
\boldsymbol{x}_i &= \boldsymbol{z}_i \quad \text{for } k < i \leq d.
\end{aligned} \tag{A.33}$$

Then we observe that

$$\sum_{i=1}^k \boldsymbol{\alpha}_i \boldsymbol{x}_i = \boldsymbol{z}_1 \tag{A.34}$$

and thus the inverse is given by

$$\begin{aligned}
\boldsymbol{z}_1 &= \sum_{i=1}^k \boldsymbol{\alpha}_i \boldsymbol{x}_i, \\
\boldsymbol{z}_2 &= \boldsymbol{\alpha}_2 \boldsymbol{x}_2 + \left(\sum_{i=1}^k \boldsymbol{\alpha}_i \boldsymbol{x}_i\right)^k, \\
\boldsymbol{z}_i &= \boldsymbol{\alpha}_i \boldsymbol{x}_i \quad \text{for } 3 \leq i \leq k \\
\boldsymbol{z}_i &= \boldsymbol{x}_i \quad \text{for } d \geq i > k.
\end{aligned} \tag{A.35}$$

Similarly, we find that the inverse of the additive map given in coordinates by

$$
\begin{aligned}
\boldsymbol{x}_1 &= \frac{1}{\boldsymbol{\alpha}_1}\left(-(-\boldsymbol{z}_1)^k - \boldsymbol{z}_1 + \sum_{i=2}^{k}\boldsymbol{z}_i\right), \\
\boldsymbol{x}_2 &= \frac{1}{\boldsymbol{\alpha}_2}(-\boldsymbol{z}_2 + (-\boldsymbol{z}_1)^k), \\
\boldsymbol{x}_i &= -\frac{1}{\boldsymbol{\alpha}_i}\boldsymbol{z}_i \qquad \text{for } 3 \leq i \leq k, \\
\boldsymbol{x}_i &= \boldsymbol{z}_i \quad \text{for } k < i \leq d.
\end{aligned}
\tag{A.36}
$$

can be written as (note $\sum_{i=1}^{k}\boldsymbol{\alpha}_i\boldsymbol{x}_i = -\boldsymbol{z}_1$)

$$
\begin{aligned}
\boldsymbol{z}_1 &= -\sum_{i=1}^{k}\boldsymbol{\alpha}_i\boldsymbol{x}_i, \\
\boldsymbol{z}_2 &= -\boldsymbol{\alpha}_2\boldsymbol{x}_2 + \left(\sum_{i=1}^{k}\boldsymbol{\alpha}_i\boldsymbol{x}_i\right)^k, \\
\boldsymbol{z}_i &= -\boldsymbol{\alpha}_i\boldsymbol{x}_i \quad \text{for } 3 \leq i \leq k \\
\boldsymbol{z}_i &= -\boldsymbol{z}_i \qquad \text{for } d \geq i > k.
\end{aligned}
\tag{A.37}
$$

Summing the two inverses in equation A.33 and equation A.37 we find that the map $\boldsymbol{g}$ given by $\boldsymbol{g}_j(\boldsymbol{x}) = 2\delta_{j2}\left(\sum_{i=1}^{k}\boldsymbol{\alpha}_i\boldsymbol{x}_i\right)^k$ is in $\mathcal{G}_{\text{enc}}$ and by permuting the indices and scaling we find that all maps of the form

$$
\boldsymbol{g}_j(x) = \delta_{jl}\left(\sum_{i=1}^{k}\boldsymbol{\alpha}_i\boldsymbol{x}_i\right)^k
\tag{A.38}
$$

are in $\mathcal{G}_{\text{enc}}$. Using Lemma A.11 stated below we infer that indeed all multinomial polynomials are in $\mathcal{G}_{\text{enc}}$ and this ends the proof in light of the Stone-Weierstrass Theorem. $\qquad\square$

The following technical but standard lemma was used in the proof of Theorem A.9.

**Lemma A.11.** *Consider the space of functions $\boldsymbol{g}_{\boldsymbol{\alpha}} : \mathbb{R}^d \to \mathbb{R}$ for $\boldsymbol{\alpha} \in \mathbb{R}^d$ given by*

$$
\boldsymbol{g}_\alpha(\boldsymbol{x}) = \left(\sum_{i=1}^{d}\boldsymbol{\alpha}_i\boldsymbol{x}_i\right)^k.
\tag{A.39}
$$

*Then the vector space generated by the functions $\boldsymbol{g}_{\boldsymbol{\alpha}}$ is the space of all $k$-homogeneous polynomials.*

*Proof.* This is a general version of the well known polarization identity, namely

$$
(\boldsymbol{x}_1 + \boldsymbol{x}_2)^2 - (\boldsymbol{x}_1 - \boldsymbol{x}_2)^2 = 4\boldsymbol{x}_1\boldsymbol{x}_2.
\tag{A.40}
$$

For completeness we sketch the full proof. Denote the generated space by $V$. Let $\phi_j(\boldsymbol{x})$ be linear functions for $1 \leq j \leq k$,

i.e., $\phi_j(\boldsymbol{x}) = \sum_{i=1}^{d} \boldsymbol{\alpha}_i^j \boldsymbol{x}_i$ for some $\boldsymbol{\alpha}_i^j$. Then using the multnomial expansion we find

$$
\sum_{(\epsilon_1,\ldots,\epsilon_k) \in \{-1,1\}^k} \left( \prod_{j=1}^{k} \epsilon_j \right) \left( \sum_{j=1}^{k} \epsilon_j \phi_j \right)^k
$$

$$
= \sum_{(\epsilon_1,\ldots,\epsilon_k) \in \{-1,1\}^k} \left( \prod_{j=1}^{k} \epsilon_j \right) \sum_{\gamma_1+\ldots+\gamma_k=k} \frac{k!}{\gamma_1! \cdot \ldots \cdot \gamma_k!} \prod_{i=j}^{k} \phi_j^{\gamma_j}
$$

$$
= \sum_{(\epsilon_1,\ldots,\epsilon_k) \in \{-1,1\}^k} \left( \prod_{i=j}^{k} \epsilon_j \right) \sum_{\gamma_1+\ldots+\gamma_k=k} \frac{k!}{\gamma_1! \cdot \ldots \cdot \gamma_k!} \prod_{j=1}^{k} (\epsilon_j \phi_j)^{\gamma_j} \qquad (\text{A.41})
$$

$$
= \sum_{\gamma_1+\ldots+\gamma_k=k} \frac{k!}{\gamma_1! \cdot \ldots \cdot \gamma_k!} \prod_{j=1}^{k} \phi_j^{\gamma_j} \prod_{j=1}^{k} \sum_{\epsilon_j \in \{-1,1\}} \epsilon_j^{\gamma_j+1}.
$$

Now the last sum is 0 for $\gamma_j$ even and 2 for $\gamma_j$ odd. So the only non-zero term corresponds to all $\gamma_j$ odd and thus $\gamma_j = 1$ for all $j$ and therefore

$$
\sum_{(\epsilon_1,\ldots,\epsilon_k) \in \{-1,1\}^k} \left( \prod_{j=1}^{k} \epsilon_j \right) \left( \sum_{j=1}^{k} \epsilon_j \boldsymbol{\alpha}_j \right)^k = 2^k k! \prod_{j=1}^{k} \phi_j \in V. \qquad (\text{A.42})
$$

Clearly this implies that the monomials $\prod_{i=1}^{d} \boldsymbol{x}_i^{\boldsymbol{\beta}_i}$ with $\boldsymbol{\beta}_i \geq 0$ and $\sum_{i=1}^{d} \boldsymbol{\beta}_i = k$ are generated by the functions $\boldsymbol{g}_{\boldsymbol{\alpha}}$ (pick $\phi_j(\boldsymbol{x}) = \boldsymbol{x}_i$ for $\boldsymbol{\beta}_i$ of the $\phi_j$). $\qquad \square$

**Architectural constraints for $d_x > d_z$.** The case $d_x > d_z$ is more challenging because the data manifold is then a submanifold and even if we know that there is a $\boldsymbol{g} \in \mathcal{G}_{\text{enc}}$ inverting $\boldsymbol{f}$ on the data manifold (i.e., $\mathcal{G}_{\text{enc}}$ is sufficiently expressive) this provides (essentially) no information about $\boldsymbol{g}$ away from $\mathcal{X} = \boldsymbol{f}(\mathcal{Z})$ and the data manifolds for different generators are unrelated. However, we can leverage the result for $d_x = d_z$ to obtain a weaker version in the general case. Here we make the additional assumption that $\mathcal{G}_{\text{enc}}$ is closed under coordinate projections in the sense that $\tilde{\boldsymbol{g}} \in \mathcal{G}_{\text{enc}}$ if $\tilde{\boldsymbol{g}}(\boldsymbol{x}) = \boldsymbol{g}(\boldsymbol{x}_I, \boldsymbol{0}_{[d_x] \setminus I})$ for some index set $I \subset [d_x]$ and $\boldsymbol{g} \in \mathcal{G}_{\text{enc}}$. Note that this is naturally satisfied for neural networks where we can remove the influence of a coordinate by zeroing its outgoing weights.

**Corollary A.12.** *Assume that $\mathcal{G}_{\text{enc}}$ is a class of encoder functions such that $\mathcal{G}_{\text{enc}}$ is closed under addition and coordinate projections and sufficiently expressive, i.e., for every $\boldsymbol{f} \in \mathcal{F}_{\text{int}}$ there is $\boldsymbol{g} \in \mathcal{G}_{\text{enc}}$ such that $\boldsymbol{g} \circ \boldsymbol{f} = \mathrm{id}$. Let $\boldsymbol{f} \in \mathcal{F}_{\text{int}}$ be such that $\boldsymbol{f}_I$ is a diffeomorphism (on its image) for some $I$ with $|I| = d_z$. Then $\mathcal{G}_{\text{enc}} \circ \boldsymbol{f}$ is dense in all continuous functions $C(K, \mathbb{R}^{d_z})$ for every compact $K \subset \mathbb{R}^{d_z}$, i.e., essentially arbitrary representations can be learned using function in $\mathcal{G}_{\text{enc}}$.*

*Proof.* Consider a set $I \subset [d_z]$. Then the restrictions $\boldsymbol{f}_I$ of functions $\boldsymbol{f} \in \mathcal{F}_{\text{int}}$ such that $\boldsymbol{f}_{I^c}(\boldsymbol{z}) = \boldsymbol{0}$ (i.e., functions that vanish in all but $d_z$ coordinates) are in bijection to functions in $\mathcal{F}_{\text{int}}$ mapping $\mathbb{R}^{d_z} \to \mathbb{R}^{d_z}$. Applying Theorem A.9 we therefore find that the set of functions $\boldsymbol{z}_I \to \boldsymbol{g}(\boldsymbol{z}_I, \boldsymbol{0}_{I^c})$ for $\boldsymbol{g} \in \mathcal{G}_{\text{enc}}$) is dense in the continuous functions defined on any compact set $K'$. It is convenient to introduce the shorthand $\bar{\boldsymbol{g}}$ for the function $\boldsymbol{z}_I \to \boldsymbol{g}(\boldsymbol{z}_I, \boldsymbol{0}_{I^c})$ by $\bar{\boldsymbol{g}}$. Then we can restate the density statement before as follows: Given any continuous function $\boldsymbol{h} : K \to \mathbb{R}^{d_z}$ we can find for any $\epsilon > 0$ a $\boldsymbol{g} \in \mathcal{G}_{\text{enc}}$ so that $\|\bar{\boldsymbol{g}} - \boldsymbol{h} \circ (\boldsymbol{f}_I)^{-1}\| < \epsilon$ on the compact set $K' = \boldsymbol{f}_I(K)$ (here we use that $\boldsymbol{f}_I$ is bijective on its image to invert it). Using that $\mathcal{G}_{\text{enc}}$ is closed under coordinate projections we can find $\tilde{\boldsymbol{g}}$ is in $\mathcal{G}_{\text{enc}}$ and satisfies

$$
\max_{\boldsymbol{z} \in K} \|\tilde{\boldsymbol{g}} \boldsymbol{f}(\boldsymbol{z}) - \boldsymbol{h}(\boldsymbol{z})\|_{\infty} = \|\boldsymbol{g}(\boldsymbol{f}_I(\boldsymbol{z}), \boldsymbol{0}_{I^c}) - \boldsymbol{h} \circ (\boldsymbol{f}_I)^{-1} \circ \boldsymbol{f}_I(\boldsymbol{z})\|_{\infty}
$$

$$
= \|\bar{\boldsymbol{g}}(\boldsymbol{f}_I(\boldsymbol{z})) - \boldsymbol{h} \circ (\boldsymbol{f}_I)^{-1}(\boldsymbol{f}_I(\boldsymbol{z}))\|_{\infty} \qquad (\text{A.43})
$$

$$
\leq \max_{\boldsymbol{x}_I \in K'} \|\bar{\boldsymbol{g}}(\boldsymbol{x}_I) - \boldsymbol{h} \circ (\boldsymbol{f}_I)^{-1}(\boldsymbol{x}_I)\|_{\infty}.
$$

This ends the proof.

$\qquad \square$

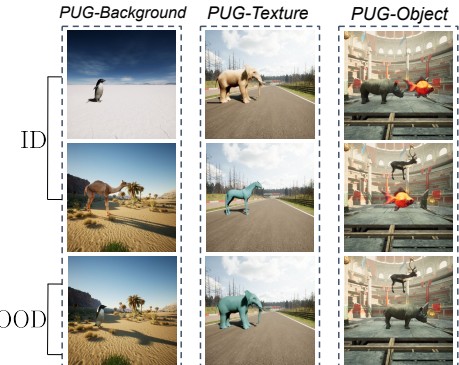

*Figure 7.* ID and OOD images from each dataset used in our experiments.

While the previous corollary makes the strong assumption that $\boldsymbol{f}_I$ is globally bijective we note that this is generally true at least locally. Moreover, we can patch such representations due to the additive structure. Therefore, it seems unlikely that a function class $\mathcal{G}_{\text{enc}}$ satisfying the following three constraints exists: First, the function class $\mathcal{G}_{\text{enc}}$ is expressive enough, i.e., it contains left inverses for all $\boldsymbol{f} \in \mathcal{F}_{\text{int}}$. Secondly, $\mathcal{G}_{\text{enc}}$ is not too expressive so it provides a useful inductive bias towards $\mathcal{G}_{\text{int}}$. And finally, $\mathcal{G}_{\text{enc}}$ can be efficiently parametrized and used for optimization.

## B. Additional Experimental details

**Datasets.** We create datasets for our experiments in Sec. 5 based on the PUG: Animals dataset (Bordes et al., 2023). This data consists of 43,520 high-resolution images which we resize to $224 \times 224 \times 3$. Images consist of two animals from 32 categories which can occupy four different positions (left, right, up, down). Each animal can take 5 different colors/textures (standard, blue, red, grass, stone). These animals are then placed on a background which can take 10 different values.

To create PUG-Background, we create an OOD set containing 32,000 images which consist of unseen combinations of animal category and backgrounds, e.g., penguin in a desert in Fig. 7, and a corresponding ID set containing 11,520 images. For PUG-Texture, the OOD set contains 16,000 images consisting of unseen combinations of animal and texture/color, e.g. blue elephant in Fig. 7, and the ID set contains 27,520 images. Lastly, for PUG-Object, the ID and OOD set both contain 21,760 images. The OOD set here consist of unseen combinations of animal categories, e.g., rhinoceros and caribou in in Fig. 7.

### B.1. Models

**ViTs.** The exact architecture for the six different pretrained ViTs used in our experiments can be seen in Fig. 5. When fine tuning these models with a LORA adapter, we use a rank of 16, a scaling factor of 32, and a dropout value of 0.1.

**Slot encoders.** We use either a Transformer or Slot Attention model for the slot encoder in Sec. 5. Both models consist of 5 layers and use 3 slots of dimension 64. We use 4 attention heads in the Transformer.

**Decoders.** All decoders in our experiments use the cross-attention Transformer from Brady et al. (2025); van Steenkiste et al. (2024); Sajjadi et al. (2022). This decoder first projects slots using a 2 layer slot-wise MLP with a hidden dimension of 2064 and then passed through a 2 layer cross-attention Transformer with pixel queries. Pixels are tokenized using a 2 layer MLP with a hidden dimension of 720. The outputs of the Transformer are mapped down to a channel dimension of 3 using a 3 layer MLP with a hidden dimension of 360. The attention weights $\boldsymbol{A}$ in the Transformer are regularized such that pixels are encouraged to attend to at most one slot using the regularizer introduced by Brady et al. (2025) weighted by a value of 0.01:

$$\mathcal{L}_{\text{interact}} := \mathbb{E} \sum_{l \in [d_x]} \sum_{j \in [K]} \sum_{k=j+1}^{K} \boldsymbol{A}_{l,j} \boldsymbol{A}_{l,k}. \tag{B.1}$$

### B.2. Training Objectives

**Supervised models.** We train all supervised models for 100000 iterations across 3 random seeds using a batch size of 64, with the Adam optimizer (Kingma & Ba, 2015) and a learning rate of 1e-4.

**VAEs.** We train all unsupervised VAE models for 300000 iterations across 3 random seeds using a batch size of 32, with the Adam optimizer (Kingma & Ba, 2015) and a learning rate of 5e-4, which is decayed by a factor of .1 throughout training and warmed up for the first 10000 iterations. We use a value of either 0.005 or 0.001 for the hyperparameter $\beta$ on the KL loss.

**Readout.** In our unsupervised experiments, we train a linear readout on learned slots for 7500 iterations. To resolve the permutation between inferred and ground-truth slots we rely on on the Hungarian matching procedure used in Dittadi et al. (2022); Locatello et al. (2020).

**Gradient-based search.** When performing gradient based search in our experiments, we optimize Eq. 4.3 using Adam with a learning rate of .001. We optimize for either 300 or 500 iterations on PUG-Background and 700 iterations on PUG-Texture. To further aid in optimization we add an additional regularizer to the optimization procedure which minimizes the entropy of the logits under the classifier. This aims to ensure that the search procedure yields latent slots which are within the set of slots which the classifier has already observed. We use a value of either 10 or 50 for this loss. We note that a similar loss was used for semi-supervised learning in Grandvalet & Bengio (2004).

**Generative replay.** For our experiments using generative replay, we generate OOD data by following the procedure in Wiedemer et al. (2024) in which ID slots are randomly shuffled to create novel OOD compositions. We train an encoder on batches of 64 OOD samples for 15000 iterations with a learning rate of 5e-4.

**Compute.** We train all models using 2 NVIDIA A100 GPUs. Total training time was approximately 1500 GPU hours.

```python
import torch
from torch.optim import Adam

def latent_search(x_ood, encoder, decoder, learning_rate, num_iters):

    # get encoding for ood image
    zh = encoder(x_ood)
    optimizer = Adam(params=[zh],
    lr=learning_rate)

    # optimize encoding to minimize mse under decoder
    for _ in range(num_iters):
        xh = decoder(zh)
        mse_loss = (x_ood - xh).square().mean()
        mse_loss.backward()
        optimizer.step()

    return zh

def generative_replay(zh_ood, encoder, decoder, learning_rate):

    optimizer = Adam(params=encoder.parameters(),
    lr=learning_rate)

    # ensure gradients not computed for decoder
    for param in decoder.parameters():
        param.requires_grad = False

    # generate OOD image
    xh_ood = decoder(zh_ood)

    # re-encode image with encoder
    zh_ood_recon = encoder(xh_ood)

    # ensure re-encoding matches original encoding
    mse_loss = (zh_ood - zh_ood_recon).square().mean()
    mse_loss.backward()
    optimizer.step()

    return encoder
```

*Figure 8.* Top. PyTorch pseudocode for gradient-based search using a decoder for a given OOD image $x$ § 4.1. Bottom. PyTorch pseudocode for one gradient step of generative replay § 4.2.

# C. Additional Experiments

In this appendix, we report results for compositional generalization when using decoders with less constrained inductive biases.

## C.1. Unconstrained Transformer Decoder

*Table 2.* ID and OOD accuracy (%) across different architectures.

| Architecture | ID ↑ | OOD ↑ |
|---|---|---|
| Constrained Decoder | 99.9 ±0.04 | 83.43 ±0.87 |
| Flexible Decoder | 99.77 ±0.05 | 78.53 ±2.37 |
| Encoder-only (Slot Attention) | 99.8 ±0.2 | 0.002 ±0.002 |
| Encoder-only (Transformer) | 100.0 ±0.0 | 0.008 ±0.006 |

**Setup.** We first train a Transformer decoder that uses self-attention between patches and removes the attention regularization. The model is trained with the same VAE loss as all prior models. We train this model on PUG-Background and report OOD results after leveraging replay.

**Results.** In Tab. 2, we observe comparable OOD performance when using this decoder compared to our more constrained Transformer decoder. This result provides evidence that generative methods may also be capable of achieving compositional generalization by relying on more implicit inductive biases from e.g. the dynamics of optimization. In contrast, we observe that non-generative methods fail to generalize OOD through such implicit mechanisms.

## C.2. Slot-based vs. non-slot-based decoder

**Setup.** In this section, we test to what extent a decoder which operates on a monolithic latent vector opposed to slots can generalize compositionally. To test this, we train an autoencoder on PUG-Background with the following structure: images are mapped by a CNN encoder to a latent representation with the same total dimensionality as our slot-based model, but which does not have a slot structure. This latent is then decoded by an CNN decoder which is not slot-based. We train this model on PUG-Background across 3 seeds using the same training settings as § 5 and with a VAE loss with $\beta = 0.005$. For the CNN encoder and decoder, we leverage the same architecture used in Dittadi et al. (2022). To evaluate classification performance, we train 3 separate linear readouts on the latents of this model to predict each animals and background. To evaluate whether this decoder offers gains in OOD performance, we perform 750 iterations of search § 4.1 under the trained decoder and re-evaluate OOD accuracy. We benchmark the performance of this unstructured model against our structured Transformer decoder from § 5 which uses a ViT-Small base encoder trained from scratch. Similarly, we perform 750 iterations of search under this decoder, and re-evaluate OOD performance.

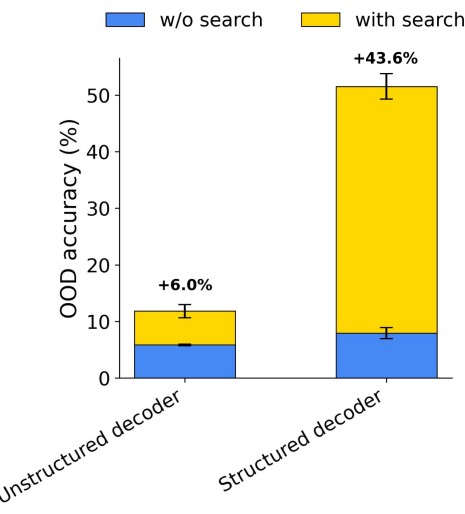

*Figure 9.* OOD accuracy on PUG-Background after applying search § 4.1 using a structured slot-based Transformer decoder designed according to § 3 vs. an unstructured CNN decoder.

**Results.** For both models, we observe near-perfect ID classification performance. Thus, similar to § 5, we only report OOD performance. Results can be seen in Fig. 9. We find that both models achieve similar OOD accuracy before leveraging search. However, after applying search, we observe significantly higher gains in OOD performance when using the structured Transformer decoder compared to the unstructured CNN. These results suggest that non-trivial gains in compositional generalization may required a slot-based structure on a decoder.

# D. Extended Discussion

**More complex datasets.** One limitation of our experimental study is that we do not evaluate compositional generalization on real-world image datasets. The main issue in leveraging such datasets is that evaluating compositional generalization requires data in which all ground-truth latent factors in each image (e.g. objects, backgrounds, textures, etc.) are explicitly known, thereby enabling us to construct ID and OOD splits in a principled way. For real-world, unstructured data the underlying latents are unknown making such datasets unsuitable for rigorously testing compositional generalization. Consequently, we elected to use PUG, which to the best of our knowledge is the most visually complex data in which the underlying latents are known. Yet, this dataset is still limited in its visual complexity relative to real world data. We thus believe that an important direction for future work is to create a large scale, visually realistic image dataset which offers access to the ground-truth latents, perhaps by leveraging a synthetic or neural network based image renderer.

**Compositional generalization via data scale.** A key result of our experiments in § 5 is that non-generative methods leveraging encoders with large-scale pretraining achieve substantial gains in OOD performance. This suggests that, as the pretraining size of the base encoder increases, the full encoder (base encoder plus slot encoder) becomes initialized in a region of the loss landscape where all reachable optima correspond to models that also generalize OOD. A limitation of our current experiments is that we cannot investigate this phenomenon in a more systematic manner, since we rely on pretrained encoders that differ in architecture, training objective, and pretraining dataset. A more principled approach would require an empirical study in which an encoder's architecture and training objective are held fixed, while the model is trained on gradually increasing amounts of pretraining data. We leave such an investigation as an important direction for future work.

**Computational cost of generative methods.** One drawback of leveraging generative methods is that training a decoder as well as using search or replay add computational overhead relative to encoder-only methods. For example, in our experiments in § 5.1, a single iteration for a supervised ViT-Small base encoder took $\sim 55$ ms for a batch size of 32, while adding a Transformer decoder and training with a VAE loss increased this time to $\sim 366$ ms. This overhead can be reduced by only decoding a subset of pixels similar to van Steenkiste et al. (2024), but we leave this for future work. Additionally, using generative replay incurs approximately $\sim 170$ ms per iteration, while search requires $\sim 215$ ms with a batch size of 32. We note that we train for only 15,000 replay iterations, making this procedure relatively computationally scalable. Search, on the other hand, requires 750 iterations per batch. Thus, further work is needed to scale this procedure more effectively.

**Limitations of assumptions in theory.** Our theoretical results in § 3, rely on two main assumptions on the ground-truth generator $f$: (i) that $f$ is a diffeomorphism and (ii) that the interactions between slots under $f$ are restricted according to Eq. (2.7). While these assumptions are the most general which have been shown to enable compositional generalization, it is possible that they may fail to accurately model real-world data in certain circumstances. For example, the assumption that $f$ is a diffeomorphism may not hold for images in which objects exhibit strong occlusions such that $f$ is no longer invertible. Additionally, it is possible that certain interactions between concepts may not be captured by Eq. (2.7) such as complex shadows or reflections.

**Limitations of assumptions in practice.** Enforcing our theoretical assumptions on a model in practice can potentially pose scalability challenges. For example, enforcing that a model is a diffeomorphism via an autoencoder is computationally costly and thus can be challenging to scale. We note, however, that modern generative diffusion models and classifiers (Rombach et al., 2022; Peebles & Xie, 2023; Jaini et al., 2024) rely on learning a diffeomorphic mapping between image and latent space, highlighting the feasibility of enforcing this assumption at scale. Furthermore, enforcing that a decoder takes the form of Eq. (2.7) exactly is not scalable in practice (Brady et al., 2025). To this end, we aim to approximate this form in our experiments using a structured Transformer decoder and restricting interactions via a KL penalty and a sparsity regularizer on the decoder's attention weights (Brady et al., 2025). Importantly, we find this model yields significant gains for compositional generalization highlighting that enforcing that a decoder exactly matches Eq. (2.7) may not be necessary in practice. Furthermore, we suspect that biologically plausible regularizers such as in Whittington et al. (2023) which aim to minimize activation energy through sparsity regularizers on the latents and the decoder's weight matrix could serve a similar role as our regularizers.

