# OpenReview forum: "Is Generation Required for Data-Efficient Perception?"
_ICML.cc/2026/Conference — ICML 2026 regular_

### Official Review · Reviewer_jSZM · 2026-03-09

**Soundness:** 2
**Presentation:** 2
**Significance:** 3
**Originality:** 3
**Overall Recommendation:** 4
**Confidence:** 2

**Summary:**

This paper analyzes from a theoretical perspective that to achieve data-efficient perception, generation methods are required, while non-generation methods usually cannot achieve this goal. In the theoretical analysis, the authors propose using the perception ability on OOD (out-of-domain) samples to measure a model's compositional generalization ability.

**Compliance With Llm Reviewing Policy:**

Affirmed.

**Final Justification:**

I am satisfied with the rebuttal and will maintain my rating.

**Key Questions For Authors:**

Please see the "Weaknesses" part.

**Limitations:**

yes

**Strengths And Weaknesses:**

Strengths:
1. This paper provides sufficient theoretical analysis, and includes corresponding mathematical modeling for the problems it discusses.
2. Regarding the data-efficient perception problem of models, this paper proposes a novel perspective by comparing two different approaches: generation and non-generation methods, to analyze the model's perception ability.

Weaknesses:
1. Presentation: This paper introduces a lot of mathematical models and theorems. However, some mathematical variables' meanings are unclear, making the paper hard to follow. For example, in the "The problem of OOD identifiability" part of Section 2 (Lines 113-123 on the right column), the paper introduces $f^1$ and $f^2$, and states that "$f^1 \circ h_\pi = f^2 \text{ on } Z_{ID}$", etc. However, what do $f^1$ and $f^2$ generate, since they are generators? And why does the author state that "$f^1 \circ h_\pi = f^2 \text{ on } Z_{ID}$"? What is the exact meaning of this formula? It would be better if the authors could clearly explain the specific meaning of each variable and formula when introducing these mathematical models, and provide relevant examples to help readers understand easily.
2. The definition of OOD images is not sufficient: According to Equation (2.4), OOD images are defined as images with unseen combinations (e.g., a penguin on a beach). However, OOD samples should also include images containing objects that the model has never seen before, rather than simply combinations of known objects/textures/backgrounds. Can the analysis in the paper be extended to a broader definition of OOD samples?

---

> ### Author Rebuttal · Authors · 2026-03-30
>
> We thank the reviewer for their time and feedback. We address each of your points below.
>
> **Comment:**
>
> “what do ${f}^1$ and ${f}^2$ generate…? And why does the author state that "$f^1 \circ h_\pi = f^2$ on ${Z}_{ID}$"? What is the exact meaning of this formula?
>
> **Response:**
>
>  $f^1$ and $f^2$ are generators which map from latent slots to images. In the context of this formula, $h_\pi$ is a slot-wise invertible function which is defined in lines 99-103. The meaning of the formula $f^1 \circ h_\pi = f^2$ on $Z_{ID}$ is that the two functions  ${f}^1$ and ${f}^2$ are equal on the set of in-distribution latents $Z_{ID}$ up to $h_\pi$, i.e., up to invertible transformations of the slots.
>
> **Comment:**
> “The definition of OOD images is not sufficient. Can the analysis in the paper be extended to a broader definition of OOD samples?"
>
> **Response:**
>
> We think there may be a misunderstanding which we aim to clarify. Our paper studies conditions under which compositional generalization is possible. Thus, our definition of OOD images is specific to this setting and does not aim to capture all possible types of OOD data one may encounter.
>
> In general, OOD generalization is challenging and requires stringent assumptions. Analysis of other types of distribution shifts is likely possible but no approach can generalize to arbitrary OOD settings, and we therefore focused on the important setting of compositional generalization..

---

> > ### Author Rebuttal · Reviewer_jSZM · 2026-04-03
> >
> > I thank the authors for their concise clarifications. The response has largely addressed my initial concerns. Here are a few minor suggestions for the revision:
> >
> > The explanation of the equation provided in the rebuttal is clear. I strongly recommend incorporating this plain-text explanation directly into the manuscript to help future readers easily follow the equations.
> >
> > Scope of OOD: I completely understand and accept the authors' clarification that the paper specifically focuses on compositional generalization. It would be beneficial to explicitly emphasize this scope boundary in the introduction or problem setting to prevent similar misunderstandings from other readers.
> >
> > Overall, I am satisfied with the rebuttal and will maintain my rating.

---

### Official Review · Reviewer_ze1e · 2026-03-13

**Soundness:** 3
**Presentation:** 3
**Significance:** 2
**Originality:** 3
**Overall Recommendation:** 4
**Confidence:** 4

**Summary:**

This paper compares generative (decoder-based) and non-generative (encoder-based) approaches to compositional generalization. It argues that, under certain assumptions, a decoder can be more easily constrained to match the desired functional form, while imposing equivalent constraints on an encoder is impossible without knowledge of the data manifold. The paper further introduces two methods for inverting a learned decoder to recover latents of OOD images with the help of an encoder. Proof-of-concept experiments on the PUG dataset show that the generative approach outperforms the non-generative one without large-scale pre-training.

**Compliance With Llm Reviewing Policy:**

Affirmed.

**Final Justification:**

The paper studies generative vs. non-generative approaches to perception tasks through the lens of data efficiency. The theoretical analysis provides useful insights that align with empirical observations and is supported by experiments. Although there remains a gap between the theory and practice, the work represents a meaningful step toward understanding the generalization capabilities of different modeling paradigms.

**Key Questions For Authors:**

- Under the current experimental setup, the authors are encouraged to report results for the decoder-based approach without relying on a pre-trained encoder, i.e., gradient-based search without encoder bootstrapping, and training an encoder from scratch on synthetic OOD data. This helps calibrate the decoder performance w.r.t. the encoder-only approach.

- Although real-world generative architectures do not strictly follow the functional form adopted by the paper, they also introduce structural bottlenecks (e.g., low-dimensional latent spaces, Gaussian priors, progressive feature upsampling). This suggests that the data efficiency argument may still hold. To strengthen the paper, the authors are encouraged to include experiments in more realistic (yet still controlled) settings. One such example is StyleGAN trained on human faces, where the generator empirically learns to factorize facial attributes in its latent space. Such experiments are well motivated by the theoretical analysis and would substantially strengthen the paper’s claim.

**Limitations:**

The paper has included an extensive discussion on the limitations and societal impact of the work.

**Strengths And Weaknesses:**

### Strength

- The paper introduces a theoretical framework for comparing encoder and decoder-based perception. Under a specific functional form of the latent image model, the authors prove that compositional generalization can be more readily enforced in a decoder-based approach. This analysis offers new insights into the structure of the perception task.

- To make the decoder-based framework practical, the paper proposes two methods for bootstrapping the inversion of OOD images given an imperfect encoder: (a) gradient-based search and (b) fine-tuning the encoder on synthetic OOD data. These approaches help bridge the gap between the theory and practice.

- The authors conduct proof-of-concept experiments to validate their theoretical results. They show that the generative approach outperforms the non-generative one when both models are trained from scratch, while the non-generative approach catches up when the encoder has undergone large-scale pre-training. These results supports the paper’s claim on data efficiency.

### Weaknesses

- The paper is heavy on theory, light on empirical validation, while making very broad claims (e.g., the title). At a high level, the data efficiency and compositional generalization of the generative approach largely stem from bottlenecks in the latent structure and model architecture, instantiated in the paper through an arguably restrictive functional form of the decoder. While the assumption simplifies the theoretical analysis, it creates a significant gap between the theory and practice, given the complex nature of perception tasks, data, and model architectures.

- Latent recovery in the decoder-based approach relies heavily on a reasonably strong encoder. The effectiveness of gradient-based search remains unclear without a good initialization given by the encoder. Similarly, generative replay assumes access to an initially in-distribution encoder and requires additional synthetic OOD data for subsequent adaptation. This dependence entangles the contributions of the encoder and decoder and weakens the claims of the paper.

- The paper only demonstrates experimental results on a toy dataset. In reality, compositional generalization is far more complicated than simple combinatorics, and it remains unclear how the proposed framework would translate to perception in real world.

---

> ### Author Rebuttal · Authors · 2026-03-30
>
> We thank the reviewer for their time and constructive feedback. We address each of your points below.
>
> **Comment:**
>
> “The paper is heavy on theory, light on empirical validation, while making very broad claims (e.g., the title).”
>
> **Response:**
>
> We appreciate the reviewer’s comment and agree that there is a gap between the scope of our results and the breadth of the claim made in the current title. In response, we propose changing the title to “Is Generation Required for Data-Efficient Perception?” We believe this better reflects the intent of the paper: not to make a definitive claim, but to investigate this question and provide evidence toward answering it.
>
> **Comment:**
>
> “The data efficiency … of the generative approach largely stems from.. latent structure and .. architecture, ... through a .. restrictive ... form of the decoder. While this simplifies the theoretical analysis, it creates a ... gap between .. theory and practice”
>
> **Response:**
>
> Thank you for raising this point. As noted also in our reply to R. n1m2, we agree that there exist a large gap between decoders used in practice and the restricted transformer architecture used in our experiments based on our theory. We attempt to close this gap by training a much more flexible decoder, designed to better resemble decoders used at scale in e.g. [(diffusion generative models)](https://arxiv.org/abs/2403.03206), and evaluated whether it still generalizes OOD. Precise details on this model and experimental setup can be found in our reply to R. n1m2.
>
> We find that this model achieves comparable OOD performance to the more constrained Transformer used in our earlier experiments. Results can be found in the table in our reply to  R. n1m2. This result provides evidence that the data-efficiency advantages of generative methods are not limited to settings with explicitly regularized decoders, but can also persist when use flexible architectures
>
> **Comment:**
>
> “Although real-world generative architectures do not strictly follow the .. form adopted by the paper, they also introduce structural bottlenecks…This suggests that the data efficiency argument may still hold...”
>
> **Response:**
>
> We appreciate this insightful comment by the reviewer and believe that our aforementioned experiments have provided evidence supporting the reviewer's intuition. Namely, that more flexible generative architectures which do not exactly match our theory also exhibit data efficiency gains. We note, however, that this does not arise across all decoder architectures. As shown in Appendix C, the CNN decoder, which possesses a weaker inductive bias, generalizes poorly OOD.
>
> **Comment:**
>
> “Latent recovery in the decoder-based approach relies heavily on a reasonably strong encoder. The effectiveness of … search remains unclear without a good initialization given by the encoder. … This dependence … weakens the claims of the paper.”
>
> **Response:**
>
> We believe there is a misunderstanding that we would like to clarify.
>
> Our claims on data efficiency concern generative approaches. The distinction between non-generative and generative approaches is not whether an encoder is used, but whether the representation is obtained by inverting a decoder (lines 142–149). In practice, an encoder can be used in generative approaches both for training the decoder effectively and for initializing gradient-based inversion at test time.
>
> However, the use of an encoder does not imply that the OOD generalization of the decoder cannot be isolated from the encoder. We believe the confusion stems from conflating two separate ideas: (1) learning a decoder that generalizes, and (2) being able to efficiently leverage that decoder for representation learning.
>
> Figure 6B helps clarify this point. The blue bar shows OOD performance using the encoder representation directly. The yellow bar starts from that same representation and refines it using decoder feedback. The gain from blue to yellow is therefore attributable to the decoder: both conditions use the same encoder-derived initialization, and the only difference is the decoder-based refinement. Thus, this improvement reflects decoder-driven generalization.
>
> If optimization from a random initialization were to fail, this would not imply that the decoder failed to generalize. Rather, it would show that its generalization is not easily exploitable without a good starting point. This is a question of optimization, not of whether the decoder itself has learned useful generalizable structure.
>
> If this clarification does not fully resolve the reviewer’s concern, we would be happy to further address this point and, if needed, include the suggested additional experiments.
>
> **Comment**:
>
> “ .. paper only demonstrates .. results on a toy dataset. it remains unclear how the .. framework would translate to .. real world.”
>
> **Response:**
>
> We refer the reviewer to our response to a similar comment from Reviewer n1m2 . Due to character constraints, we do not repeat it here.

---

> > ### Author Rebuttal · Reviewer_ze1e · 2026-04-04
> >
> > Thanks for the rebuttal. It has fully addressed my concerns.

---

> > > ### Author Response · Authors · 2026-04-05
> > >
> > > We are pleased to hear that we have fully addressed the reviewer’s concerns.
> > >
> > > Given this, we would like to ask whether the reviewer was considering raising their score to more confidently recommend acceptance, or if there are any further points we could clarify to facilitate that decision.

---

### Official Review · Reviewer_HZtq · 2026-03-13

**Soundness:** 2
**Presentation:** 2
**Significance:** 2
**Originality:** 3
**Overall Recommendation:** 3
**Confidence:** 4

**Summary:**

This paper argues that generative methods are essential for vision models to achieve compositional generalization on out-of-distribution datasets. The authors theoretically demonstrate that generative approaches can structurally enforce the inductive biases necessary for such generalization by constraining and inverting a decoder, whereas enforcing similar structural constraints on encoder-based methods is generally infeasible, as it requires prior knowledge of the geometric structure of the OOD data manifold, which is typically unknown. The paper empirically evaluates both generative and non-generative approaches on the PUG dataset, claiming that generative methods are more data-efficient in achieving compositional generalization on OOD compositions.

**Compliance With Llm Reviewing Policy:**

Affirmed.

**Final Justification:**

Thanks to the authors' clarification that decoder inversion itself provides clear value over encoder-only approaches. However, a tension remains between the theoretical claims and the empirical evidence. The paper positions the $F_{int}$ constraints as the essential mechanism distinguishing the proposed approach from data augmentation. Yet the rebuttal shows that a flexible Transformer decoder without strict $F_{int}$ constraints achieves 78.53% OOD accuracy compared to 83.43% for the constrained version, suggesting that comparable performance can be obtained without the proposed structural constraints. This raises questions about whether the theoretical framework is necessary to explain the observed gains. Instead, the improvements may largely stem from the slot-based architecture and the generation process itself, making it unclear how the approach fundamentally differs from generation-based data augmentation.
In addition, validation remains limited to a single synthetic dataset with a relatively small combinatorial space, which makes it difficult to assess the generality of the claims.
Overall, these concerns remain unresolved, and I would maintain my score of 3.

**Key Questions For Authors:**

-  Are there experimental results on benchmarks other than PUG, such as CLEVR-based datasets? Without comparisons on benchmarks commonly used in the compositional generalization literature, it is difficult to assess the relative performance of the proposed method.
- Despite data efficiency being explicitly stated in the title, the paper provides no quantitative comparison of how much data each approach requires to reach equivalent OOD performance. Can the authors provide concrete numbers quantifying the data efficiency gap between generative and non-generative methods?
- Is there an ablation studying how OOD performance changes as the number of replay samples is reduced? Without such an ablation, it is difficult to determine whether the encoder learns genuine compositional generalization or simply fits the synthetic distribution generated by the decoder.
- Is there an ablation on the number of gradient-based search iterations? An analysis of the performance trade-off against computational cost would allow for a clearer assessment of the practical applicability of the proposed approach.

**Limitations:**

The authors openly discuss several limitations of their work, including the restriction of the theory to the F_int function class, the potential failure of Eq. 2.7 to capture complex concept interactions such as occlusion, the limited visual complexity of the PUG dataset, and the computational cost of generative methods. However, two important limitations are not addressed. First, generative replay generates approximately 960,000 samples from the same combination space as the 32,000 image OOD test set, potentially saturating the OOD test distribution prior to evaluation. Second, the asymmetric inference computational cost between gradient-based search and non-generative baselines raises concerns about the fairness of the experimental comparisons. Additional discussion of these two limitations is needed.

**Strengths And Weaknesses:**

### Strengths
- This paper provides a theoretical formalization of why enforcing the necessary inductive biases for compositional generalization is fundamentally difficult for encoder-based methods. Theorem 3.2 meaningfully justifies the limitations of non-generative methods that had previously only been observed empirically.
- Based on this theoretical analysis, the paper proposes two concrete strategies for OOD decoder inversion, gradient-based search and generative replay, representing a notable attempt to bridge the gap between theory and practice.
- The direct comparison between a structured Transformer decoder and an unstructured CNN decoder in Appendix C empirically demonstrates that structural constraints on the decoder are critical for OOD performance gains, concretizing what kind of generative approach is necessary.

### Weaknesses
- The theoretical claims are limited to generators belonging to F_int, leaving complex real-world cases such as occlusion unaddressed.
- The paper lacks quantitative evidence to support the data efficiency claimed in the title. Without a direct comparison of the amount of training data required by generative and non-generative methods to achieve equivalent OOD performance, the degree of data efficiency remains unclear.
- While the authors acknowledge in Appendix D that decoder training, search, and replay all introduce significant computational overhead, the paper provides no ablation on whether comparable OOD performance can be achieved with fewer search iterations, making it difficult to assess practical applicability.
- The experiments rely solely on the synthetic PUG dataset, leaving it unclear whether the proposed approach generalizes to more complex real-world settings.
- The non-generative baselines are implemented internally and are not compared against established SOTA methods in the compositional generalization literature. The absence of comparisons on commonly used benchmarks such as CLEVR makes it difficult to contextualize the proposed method's performance.
- The OOD test set for PUG-Background contains 32,000 images, yet generative replay generates approximately 960,000 samples from the same combination space, roughly 30 times the size of the OOD test set. Without an ablation on the number of replay samples, it is impossible to distinguish genuine compositional generalization from memorization of the decoder-generated distribution.

---

> ### Author Rebuttal · Authors · 2026-03-30
>
> We thank the reviewer for their time and detailed feedback. We address each of your points below.
>
> **Comment**
>
> “replay .. saturates the OOD . distribution ..” ..  “It is impossible to distinguish .. compositional generalization from memorization of .. generated distribution.”
>
> **Response**
>
> We believe this comment reflects confusion about compositional generalization in a generative setting.
>
> In a generative setting, the decoder's role is to learn the structure present in the training set and use it to generate novel OOD combinations. These synthetic samples are then used to train the encoder so that it can generalize to the OOD set.
>
> Thus, a decoder whose generations broadly cover, or even saturate, the OOD combination space is not a flaw. Rather, this is precisely the goal. It allows the encoder to be trained on a broad range of OOD combinations without exposing either model to the OOD test set itself.
>
> Crucially, this is not “memorization” of the OOD set, as neither the decoder nor encoder observe samples from the OOD set during training. Rather, it is compositional generalization realized by the decoder and transferred to the encoder through replay.
>
> **Comment**
>
> “The paper lacks .. evidence to support .. data efficiency claimed in . title.”
>
> **Response**
>
> We respectfully disagree. In Fig. 6, all methods are trained on exactly the same dataset. Generative methods (green/yellow) substantially improve OOD performance over non-generative (blue), providing direct evidence for better data efficiency.
>
> **Comment**
>
> "Is there an ablation of OOD performance .. as .. number of replay samples is reduced?"
>
> **Response**
>
> We added this ablation for ViT-small (see [Fig](https://drive.google.com/file/d/1ynpnYFRTfW9P_Htla5jmnNgPYHF4apuk/view)). Results show performance roughly saturates after about 2k iterations, whereas the manuscript reports 15k. Similar OOD performance can therefore be achieved with substantially less compute.
>
> **Comment**
>
> "Is there an ablation on the number of .. search iterations?"
>
> **Response**
>
> We have added this ablation (see [Fig](https://drive.google.com/file/d/1zm3rwr2EMF9qdjIFz7TSupRPHnjhGmgR/view?usp=sharing)). Performance plateaus at ~400 iterations which takes ~2.6 seconds for a batch of 16. This speedup comes from using a decoder with a larger patch size which we discuss in our reply to R. n1m2. We believe this may be further scaled by decoding only a subset of pixels and using lower precision.
>
> **Comment**
>
> “The non-generative baselines .. are not compared against established SOTA methods in the compositional generalization literature”
>
> **Response**
>
> We respectfully disagree. We evaluate a broad set of SOTA non-generative methods, including DINO, I-JEPA, CLIP, and SigLIP, using both Transformer and Slot Attention-based architectures. To our knowledge, this spans the main scalable non-generative approaches relevant to compositional generalization.
>
> **Comment**
>
> “the asymmetric .. cost between .. search and non-generative baselines raises concerns about .. fairness of the .. comparisons.”
>
> **Response**
>
> We agree that search adds compute relative to non-generative baselines, but do not view this as unfair. A key advantage of the generative approach is precisely the ability to trade additional inference-time compute for improved generalization; non-generative methods lack a comparable mechanism.
>
> **Comment**
>
> "The theoretical claims are limited .. leaving complex real-world cases such as occlusion unaddressed"
>
> **Response**
>
> We agree the theory does not exactly model effects such as occlusion, but do not see this as a major limitation. As with any theoretical analysis, some abstraction is necessary to make rigorous study possible. At the same time, we emphasize that our assumptions define the most flexible theoretical setting available in the literature for studying compositional generalization. In our view, the question is therefore not whether the assumptions perfectly capture every aspect of natural data, but whether the theory is predictive beyond its exact formal scope. Our experiments, together with the additional results in our reply to R. n1m2 using flexible decoders outside the theoretical regime, suggest that the theory’s predictions hold beyond the exact theoretical setting.
>
> **Comment**
>
> "Are there .. results on .. CLEVR? Without comparisons on benchmarks commonly used in the compositional generalization literature, it is difficult to assess .. proposed method"
>
> **Response**
>
>  We did not evaluate on CLEVR because its concepts exhibit little or no interaction, making meaningful comparison between generative and non-generative methods difficult. We also note that PUG is in fact commonly used in the compositional generalization literature; e.g., ([p1](https://arxiv.org/abs/2507.07102), [p2](https://arxiv.org/abs/2502.14113)).
>
> **Comment**
>
> "The experiments rely solely on the synthetic PUG dataset, .."
>
> **Response**
>
> We refer to our response to a similar comment from R. n1m2.

---

> > ### Author Rebuttal · Reviewer_HZtq · 2026-04-04
> >
> > I thank the authors for their thoughtful responses and additional experiments. The ablation study on the number of replay samples shows that performance saturates even with approximately 2,000 iterations, and the ablation results on search iterations substantially alleviate concerns regarding computational efficiency. In addition, it is impressive that similar OOD performance is achieved with a more flexible decoder architecture.
> >
> > Nevertheless, several key concerns remain.
> > First, there is an issue regarding the alignment between the theoretical framework and the empirical contributions. The paper claims that perception via decoder inversion is the core mechanism underlying compositional generalization. However, the experimental results suggest that most of the performance gains arise from the replay process, while the additional contribution from search appears relatively limited. Although search alone leads to some performance improvement, its magnitude is still smaller compared to that of replay. While I agree that the structural constraints of the decoder play an important role in OOD generalization, it appears that the benefits primarily stem not from the decoder inversion process itself, but rather from encoder retraining using generated data. This raises a concern about a gap between the theoretical claims of the paper and the actual mechanism driving the performance improvements.
> > Second, there are concerns regarding scalability to real-world settings. While I acknowledge that the PUG dataset serves as a useful benchmark for studying compositional generalization, the interactions between concepts in this dataset are relatively simple and may not fully capture the complexity of natural images. The additional experiments with a more flexible decoder presented in the rebuttal are encouraging; however, they are still confined to the same dataset. Without validation on more complex and realistic data, it is difficult to assess the generality of the claims made in this paper.
> >
> > Overall score: 3

---

> > > ### Author Response · Authors · 2026-04-04
> > >
> > > We thank the reviewer for their thoughtful comments and for engaging with our rebuttal.
> > >
> > > Regarding the first concern, we believe there is a key misunderstanding that we would like to clarify. Replay is not separate from decoder inversion; rather, *it is one concrete mechanism for achieving it* (Sec. 4, 4.2). Specifically, replay generates OOD samples $\hat{\boldsymbol{x}}$ using the decoder via $\hat{\boldsymbol{x}} = \hat{\boldsymbol{f}}(\hat{\boldsymbol{z}})$ and then trains the encoder $\hat{\boldsymbol{g}}$ to map $\hat{\boldsymbol{x}}$ back to the latent $\hat{\boldsymbol{z}}$ that generated it by minimizing $\| \hat{\boldsymbol{g}} (\hat{\boldsymbol{x}}) -  \hat{\boldsymbol{z}} \|$. Thus, this objective is precisely designed to enforce decoder inversion on OOD samples. Replay is therefore just as much of a mechanism for decoder inversion as search; they are simply two different approaches to the same underlying problem.
> > >
> > > Regarding the concern about scalability to real-world settings, we agree that such experiments are important for assessing the practical generality of our results. As we noted previously, however, going beyond the complexity of PUG is currently difficult because there is a lack of real-world compositional generalization datasets with controlled ID/OOD splits. Developing such datasets is something we are very interested in as future work. At the same time, we emphasize that, although our experiments focus on PUG, they already provide a non-trivial empirical signal, and the paper also makes a substantial theoretical contribution.
> > >
> > > Based on these clarifications, we respectfully ask that the reviewer reconsider their weak reject assessment.

---

### Official Review · Reviewer_n1m2 · 2026-03-17

**Soundness:** 4
**Presentation:** 3
**Significance:** 3
**Originality:** 4
**Overall Recommendation:** 5
**Confidence:** 3

**Summary:**

This paper gives theoretical support for the idea that compositional generalization in visual perception is easier to enforce through a generative route, meaning by **learning and inverting a decoder, rather than by learning an encoder directly.**

The lens of the paper is compositional generalization: in the simplest case, seeing A and seeing B should tell us something about the unseen combination of A and B. The paper argues that this kind of generalization cannot come from ID training data alone, because the desired OOD behavior is not specified by the dataset. So the issue is really about **which model can we impose the inductive bias more effectively.**

Their main theoretical point is that this structure is much easier to impose on the **decoder** than on the **encoder**. For the encoder, you only train on ID samples, so nothing in the data tells you what the encoder should do on unseen OOD inputs. In that sense, the encoder is largely left to implicit regularization and optimization luck, whereas the decoder at least admits meaningful explicit structural constraints. Albeit, there is also limit to our ability to specify the structure in the architectural sense.

On the contrary, for the decoder, the desired compositional structure can be written directly **by restricting how latent slots are allowed to interact.** This is a strong modeling assumption following Eq. 2.7 (the largest class currently shown to satisfy their identifiability condition, but not necessarily the largest possible class possible). This is both a feature and also a limitation because many realistic decoder models (Transformers, CNNs) fall outside that class, so the practical applicability of the theorem depends on how robust the story is once this assumption is violated.

Empirically, the paper trains encoder-decoder models with a structured decoder on the synthetic PUG benchmark and evaluates the learned slot representations with linear readouts. One important result is that simply training with a regularized decoder is **not enough** (follows the theoretical intuition): the encoder still only sees ID data, so using it directly at OOD test time gives only limited improvement. The bigger gains come when the paper actually **obtaining the encoder via inverting the decoder** (which is regularized) at OOD time, either by **search**, which optimizes for a latent code that reconstructs the OOD image under the decoder, or by **replay**, which generates unseen images via latent combinations which can be used to further train the encoder. These procedures substantially improve OOD performance, with replay in particular showing large gains, although all experiments are still on relatively clean synthetic data where object interactions are much simpler than in the real world.

**Compliance With Llm Reviewing Policy:**

Affirmed.

**Final Justification:**

The rebuttal especially the new OOD performance of the flexible decoder is really interesting.
I think this paper's hypothesis should be put out there to spark the discussions among the ML community.

**Key Questions For Authors:**

- **Will the encoder–decoder asymmetry remain practically meaningful in interaction-rich datasets where explicit regularization starts to hurt the performance?**

    Why? In practice, strong structural regularization is often not used because it may hurt flexibility or performance, so both encoder-based and decoder-based methods may end up relying heavily on implicit regularization.

- **How scalable are search and replay beyond controlled synthetic benchmarks?**

    Why? Search appears expensive at inference time, and replay assumes the decoder can generate meaningful unseen combinations.

**Limitations:**

Yes

**Strengths And Weaknesses:**

## Strengths

- I like that this paper asks a genuinely interesting question about the compositionality generalization.
- I find the main theoretical framing very non-trivial and insightful. The way I read the paper is: if the property we want (compositionality generalization) is external to the dataset, then it has to come from inductive bias rather than training supervision, and the real question becomes which model class lets us impose that bias more effectively.
- The paper makes a well-supported case that this is easier on the **decoder** side than on the **encoder** side.
    - For the encoder, you only train on ID samples, so nothing in the data tells you what it should do on unseen OOD inputs.
    - So encoder OOD behavior is largely left to implicit regularization and optimization luck.
    - For the decoder, at least in principle, one can directly specify structural assumptions on how latent slots are allowed to interact.
- I think this asymmetry is nontrivial and worth pointing out. Even if one does not fully buy the practical story, the theoretical point itself is interesting.
- I also like that the experiments reflect the theory which helps building the credibility of the theory.
    - Simply training an encoder-decoder model with a structured decoder does **not** automatically solve OOD generalization.
    - This matches the paper’s own logic: the encoder is still only trained on ID data, so it remains the bottleneck.
- The paper is strongest when it shows that the encoder is the bottleneck and proposes **search** and **replay** to obtain the inverse of the decoder instead.
    - Search bypasses the encoder by directly optimizing for a latent code under the decoder.
    - Replay uses the decoder to generate unseen combinations and then further trains the encoder on those.
- I can imagine that the paper sparks a meangingful discussion in the community.
    - Even outside the exact theorem assumptions, the results suggest that using a decoder at inference time may help OOD behavior more broadly (Figure 9, on unregularized CNN).
    - At the same time, the paper does not fully rule out the role of implicit regularization or strong pretrained encoders, which makes the story more complete and nuanced.
- Overall, I think the paper raises an important modeling question and gives a thoughtful answer, even if I have reservations about how far that answer extends.

## Weaknesses

- The main weakness of the paper is the strength of the modeling assumption behind Eq. 2.7 which defines a fairly restricted class of cross-slot interactions. This looks much weaker than the kinds of interactions we expect in realistic deep image models, e.g. Transformers and CNNs.
- Because of that, many realistic architectures such as Transformers and CNNs are not naturally restricted to this class.
    - So the practical relevance of the theorem depends heavily on how robust the story is once this assumption is violated. I do not think the paper fully resolves that issue.
- In the attempt to be true to the theory, the paper needs to design a decoder that: Line 303, “pixels are discouraged from attending to more than one slot”. While this strong regularization doesn’t seem to be a problem for the experiments on the synthetic datasets. It’s likely will be for more complex real-world datasets.
- In this sense, the synthetic benchmark is very aligned with the theory. PUG is clean, controlled, and has relatively simple interactions. That makes it harder to judge how much of the result would survive in realistic visual settings. It’s nice that to show that theory is correct, but it’s still very far from being practically applicable.
- More broadly, I think the practical asymmetry between encoder and decoder may be weaker than the paper suggests.
    - In modern large-scale modeling, we usually do **not** impose strong explicit structural regularization because we worry it may hurt performance.
    - So in practice, decoder-based methods also often rely on implicit regularization.
    - If that is the regime we care about, then the difference between encoder and decoder may be much smaller than the theory suggests.
- Scalability to more interaction rich datasets is also a real open question.
    - How expensive is search?
    - Can replay being able to generate meaningful unseen combinations?
    - Will the decoder be able to fit the data well enough to benefit from the OOD performance.
- I also think the presentation could be better.
    - The paper is understandable (on a second read), I wish there was an intuitive explanation on how generally the theorem goes.
    - The key conceptual move that OOD representation should come from inverting the decoder rather than trusting the encoder took me some work to digest.
    - I think the paper would benefit from a more direct intuition early on, before the formal machinery.

---

> ### Author Rebuttal · Authors · 2026-03-30
>
> We thank the reviewer for deeply engaging with our work and for recognizing the merits of our contributions. We are also grateful for the constructive feedback. We discuss each point below:
>
> **Robustness of results when theory is violated**
>
> The reviewer raised several points on the robustness of the data efficiency of generative methods when the theoretical assumptions are violated. For example:
>
> * “the practical relevance of the theorem depends … on how robust the story is once .. assumption (Eq. 2.7) is violated…"
>
> * “We usually do not impose strong … (decoder) regularization because we worry it may hurt performance.”
>
> * “Will the decoder be able to fit the data well enough to benefit from the OOD performance.”
>
> As we understand, these comments can be summarized as follows:
>
> *Our theory shows that decoders can leverage explicit regularization for compositional generalization while encoders must rely on implicit regularization. For large-scale settings, however, imposing explicit constraints on decoders may hurt performance, making implicit regularization necessary. It remains unclear whether generative methods maintain data efficiency advantages over non-generative when both methods must rely on implicit regularization.*
>
> This is a very insightful point. To address it, we trained a much more flexible decoder, designed to better resemble decoders used at scale, and evaluated whether it still generalizes OOD despite relying on implicit regularization.
>
> Namely, we removed the attention regularization, increased the patch size from 1 to 16, and added self-attention so that each layer performs both self- and cross-attention. This model is highly flexible and is similar to the architectures used in diffusion generative models [at scale](https://arxiv.org/abs/2403.03206).
>
> We trained this model on PUG-Background and then leveraged replay and search. We found that the model achieves comparable OOD performance to the more constrained Transformer used in our earlier experiments. This can be seen in the table below:
>
> | Architecture         | ID                    | OOD                  |
> |----------------------|-----------------------|----------------------|
> | Constrained Decoder  | 99.9&nbsp;±&nbsp;0.04 | 83.43&nbsp;±&nbsp;0.87 |
> | Flexible Decoder         | 99.77&nbsp;±&nbsp;0.05 | 78.53&nbsp;±&nbsp;2.37 |
>
> This provides evidence that the data-efficiency advantages of generative methods are not limited to explicitly regularized decoders, but can persist when both generative and non-generative approaches use flexible architectures and must rely on implicit regularization. We also note that this implicit regularization does not arise across all decoder architectures. As shown in Appendix C, the CNN decoder, which possesses a weaker inductive bias, generalizes poorly OOD.
>
> We will incorporate these points and add the new results in a revised version of our manuscript.
>
> **Comment:**
>
> “PUG is clean, controlled,.... That makes it harder to judge how much of the result would survive in realistic … settings”
>
> **Response:**
>
> We agree with the reviewer that scaling to more realistic datasets is important. The challenge is that evaluating compositional generalization requires precise control over the ID and OOD splits. This level of control is generally only possible with synthetically generated datasets. We thus used PUG, which is the most complex dataset of this kind that we are aware of. However, developing more complex datasets which allow controlled ID/OOD splits is an important direction for future work. We would also like to note that, although PUG is simpler than real-world data, it is still notable that non-generative methods struggle with OOD generalization even here.
>
> **Comment:**
>
> "How scalable is search and replay ...?"
>
> **Response**:
>
> Regarding replay, results from large-scale diffusion models suggest that scaling generation while preserving compositionality may be feasible. However, the extent to which these models truly generalize remains unclear, since we do not have precise knowledge of their training distributions.
>
> Regarding search, performing 400 iterations of search using our flexible decoder architecture on an NVIDIA A100 40 GB takes ~2.6 seconds. We believe this can be scaled further by decoding a fraction of pixels, and using lower numerical precision.
>
> We leave a deeper investigation of scaling replay and search for future work.
>
> **Comment:**
> “I also think the presentation could be better…”
>
> **Response:**
>
> Thank you for this thoughtful feedback! We agree that more intuition would be beneficial.
> Thus, we plan to:
> * add a few sentences in the introduction to give a more direct intuition of why non-generative compositional generalization is more challenging;
> * include a brief roadmap at the start of Section 2 to explain the structure of the section and definitions;
> * add intuition at the beginning of Section 3.1 to motivate the theoretical approach before presenting the theorems and lemmas.

---

> > ### Author Rebuttal · Reviewer_n1m2 · 2026-04-04
> >
> > The new flexible-decoder results are interesting. However, I was a bit confused here: the authors say that the advantage persists with a flexible decoder, but Appendix C seemed to conclude that meaningful gains require constraining the decoder structure. I understand that there are architectural differences, since Appendix C is CNN-based. Even so, if the authors want to use these new results to support their claim, they should at least report the encoder-only baseline as well. Otherwise, the encoder results might also improve by a similar degree, in which case the improvement would just come from changing the architecture rather than from the proposed mechanism. If so, that would not really count as positive evidence for the paper’s main claim.

---

> > > ### Author Response · Authors · 2026-04-04
> > >
> > > We appreciate the reviewer engaging with our rebuttal. We address each of your comments below.
> > >
> > > **Comment**:
> > > “ authors say that the advantage persists with a flexible decoder, but Appendix C seemed to conclude that meaningful gains require constraining the decoder structure”
> > >
> > > **Response**:
> > >
> > > We agree that this can be confusing and appreciate the opportunity to clarify. The key distinction between these decoders is not simply CNN vs. Transformer, but rather **slot-based vs. non-slot-based decoders**.
> > >
> > > Specifically, the Transformer decoders we consider, both constrained and flexible, decode latent slots, where each slot should represent an individual concept/object. This gives the decoder a direct mechanism for representing objects and a natural way to control their interactions during decoding via the attention matrix, either through explicit means (regularization, removing self-attention) or implicit means.
> > >
> > > In contrast, the CNN decoder in Appendix C operates on a **single monolithic latent vector**. To generalize compositionally, the model must then implicitly (i) organize its latent space into slot-like subspaces and (ii) learn to control interactions between them during generation. Both appear substantially more difficult without the appropriate inductive bias.
> > >
> > > Based on this, our hypothesis is that compositional generalization requires decoders that operate on slot-like latents and provide a natural mechanism for controlling interactions between them. However, once this inductive bias is present, it may not be necessary to explicitly constrain those interactions through mechanisms such as explicit regularization or removing self-attention. Instead, a flexible Transformer decoder may already provide sufficient structure for implicit regularization to enforce appropriate slot interactions, thereby enabling compositional generalization.
> > >
> > > **Comment**:
> > > “if the authors want to use these new results …, they should … report the encoder-only baseline .… Otherwise, the encoder results might also improve by a similar degree,…”
> > >
> > > **Response**:
> > >
> > > We thank the reviewer for this suggestion. We would like to clarify that, although we did not include the encoder-only baselines in the rebuttal table, these results are already reported in the main text (Table 1, first two rows of the PUG-Background results).
> > >
> > > In this setting, we train encoder-only models, mapping images to slots, with a supervised objective. To the best of our knowledge, the only scalable way to map image patch tokens into a set of latent slots is through cross-attention. We thus consider two natural cross-attention based architectures: Slot Attention (cross-attention without self-attention) and a Transformer (cross-attention with self-attention). This yields encoder-only baselines with inductive biases comparable to those explored on the decoder side.
> > >
> > > We find that both encoder-only architectures still fail to generalize OOD. This suggests that the gains reported when using a decoder are not simply due to changing the architecture. Rather, under comparable slot-based inductive biases, it appears easier for the decoder to recover the relevant structure through implicit regularization than for the encoder.
> > >
> > >
> > > To make this comparison clearer, we have now included the full table below with the encoder-only baselines alongside the new results.
> > >
> > >
> > >
> > > | Architecture                    | ID                | OOD               |
> > > |---------------------------------|-------------------|-------------------|
> > > | Constrained Decoder             | 99.9 ± 0.04       | 83.43 ± 0.87      |
> > > | Flexible Decoder                | 99.77 ± 0.05      | 78.53 ± 2.37      |
> > > | Encoder-only (Slot Attention)   | 99.8 ± 0.2        | 0.002 ± 0.002     |
> > > | Encoder-only (Transformer)      | 100.0 ± 0.0       | 0.008 ± 0.006     |

---

### Decision · Program_Chairs · 2026-04-30

**Decision:**

Accept (regular)

**Comment:**

The reviewers are leaning towards acceptance of this work. The reviewers found the theoretical aspect of the work to be interesting and valuable to the community. There were some concerns regarding the notation, which were addressed in the rebuttal. Beyond the theory, the reviewers were a bit split on the empirical results. Some find the title and validation to be a bit overclaiming, while others view it as reasonable. During the rebuttal, the reviewers mainly found their concerns to be addressed, e.g., the new validations. While there may be some gaps between the theory and practice, the AC believes the contribution outweighs the issue. The AC encourages the authors to mention that the validation is on synthetic data earlier in the paper and adjust their tone a bit in terms of the significance of the theory to have a more accurate portrayal of the content.